ന nature communications

# Genetic analysis of blood molecular phenotypes reveals common properties in the regulatory networks affecting complex traits

We evaluate the shared genetic regulation of mRNA molecules, proteins and metabolites derived from whole blood from 3029 human donors. We find abundant allelic heterogeneity, where multiple variants regulate a particular molecular phenotype, and pleiotropy, where a single variant associates with multiple molecular phenotypes over multiple genomic regions. The highest proportion of share genetic regulation is detected between gene expression and proteins (66.6%), with a further median shared genetic associations across 49 different tissues of 78.3% and 62.4% between plasma proteins and gene expression. We represent the genetic and molecular associations in networks including 2828 known GWAS variants, showing that GWAS variants are more often connected to gene expression in trans than other molecular phenotypes in the network. Our work provides a roadmap to understanding molecular networks and deriving the underlying mechanism of action of GWAS variants using different molecular phenotypes in an accessible tissue.

Genome-wide association studies (GWAS) can explain how genetic variation contributes to phenotypic variation by associating particular genomic regions to a trait of interest, often with the underlying assumption that one or multiple genes mediate this association. However, identifying the mediating molecules is still a challenge, as a large proportion of GWAS variants are located in noncoding regions with no obvious gene target[1]. Molecular studies at the population level can be used to identify these mediating molecules, by testing whether the genetic variant associated with a complex trait is also associated with gene expression levels. These expression quantitative trait loci (eQTLs) studies have been very successful in identifying molecular regulatory regions and candidate genes[2,3]. However, to understand the full causal relationship between genetic variants and complex traits, and the various stages appropriate for clinical intervention, we require studies with deep molecular phenotyping.

Recent studies have focused on molecular regulatory processes associated to GWAS studies affecting the abundance of phenotypes other than mRNA expression, such as circulating metabolites[4], plasma proteins[5–7] or other molecular phenotypes[8,9]. However, these studies often focus on one additional type of molecular phenotype, revealing only a few elements of the complete molecular path between a GWAS variant and disease. The few studies that have explored the relationships between multiple phenotypes derived from samples from the same individuals are often limited in sample size for genetics analyses[10]. A challenge for these types of study is to integrate genetic associations across molecular phenotypes to understand the downstream consequences of genetic perturbations and their cascade effects through different layers of regulation. Integrated information from these studies will be key to understand how the relationships between multiple perturbations and phenotypes define complex trait variability[11].

An additional challenge for employing molecular phenotypes to identify the full causal relationship between genetic variants and complex traits is the availability of samples from the relevant disease tissue or cell type. Our own work has previously shown that using eQTL analyses to identify genes mediating GWAS activity benefits greatly

✉e-mail: emmanouil.dermitzakis@unige.ch; ana.vinuela@newcastle.ac.uk

from data in a disease-relevant tissue[3]. For example, it took a moderately large gene expression study in pancreatic islets to detect evidence that the gene *TCF7L2* mediates the activity of the type 2 diabetes (T2D) loci with the same name. However, and in agreement with other publications[9,12], we also observed that many genetic effects are often shared across tissues, allowing to some extent the use of proxy tissues to study the genetics of common diseases. Given the practical difficulties of obtaining multiomics datasets from nonaccesible tissues, a question that remains unanswered is how deep molecular phenotyping in accessible tissues such as blood may aid in understanding the genetics of complex diseases.

Here, we use data from the DIabetes REsearCh on patient straTification (DIRECT) consortium to investigate the genetic regulation of multiple molecular phenotypes. Using blood and plasma samples from DIRECT participants, we performed local (cis) and distal (trans) quantitative trait loci (QTLs) analysis in derived gene expression, protein and metabolite phenotypes. A deep analysis of these QTLs investigated allelic heterogeneity and pleotropic effects shared across molecular phenotypes with local and distal effects. Comparison with genetic associations across multiple tissues allowed us to evaluate the value of deep phenotyping in an accessible tissue to understand the genetics of complex traits. Visualization and characterization of

regulatory networks from shared QTLs across phenotypes was used to connect clusters of genetic regulation across phenotypes and to GWAS signals. Our work has implications for the mechanistic understanding of the activity of GWAS variants, the identification of therapeutic targets for complex genetic diseases treatment and the general principles of genetic regulation of molecular phenotypes and complex traits.

## Results

### Identification of local genetic regulation of molecular phenotypes

The DIRECT dataset consists of 3029 individuals of European descent, with genotype information, gene expression quantified by RNA sequencing (RNA-seq), targeted proteomics (multiplexed immunoassays) and metabolites (targeted and untargeted mass spectrometry) in whole blood (Fig. 1A, Methods, Supplementary Table 1). To identify independent local (cis-) genetic effects that regulate mRNA expression (cis-eQTLs) and levels of circulating protein (cis-pQTLs), we performed QTL analyses followed by a stepwise regression conditional analysis for each phenotype. We identified independent significant associations for 94.4% (15,305) of genes and 97.3% (363) of proteins (Supplementary Table 2, Methods, Supplementary Data 1–3), finding similarities in the regulation of both gene expression and proteins (Fig. 1B–E, and

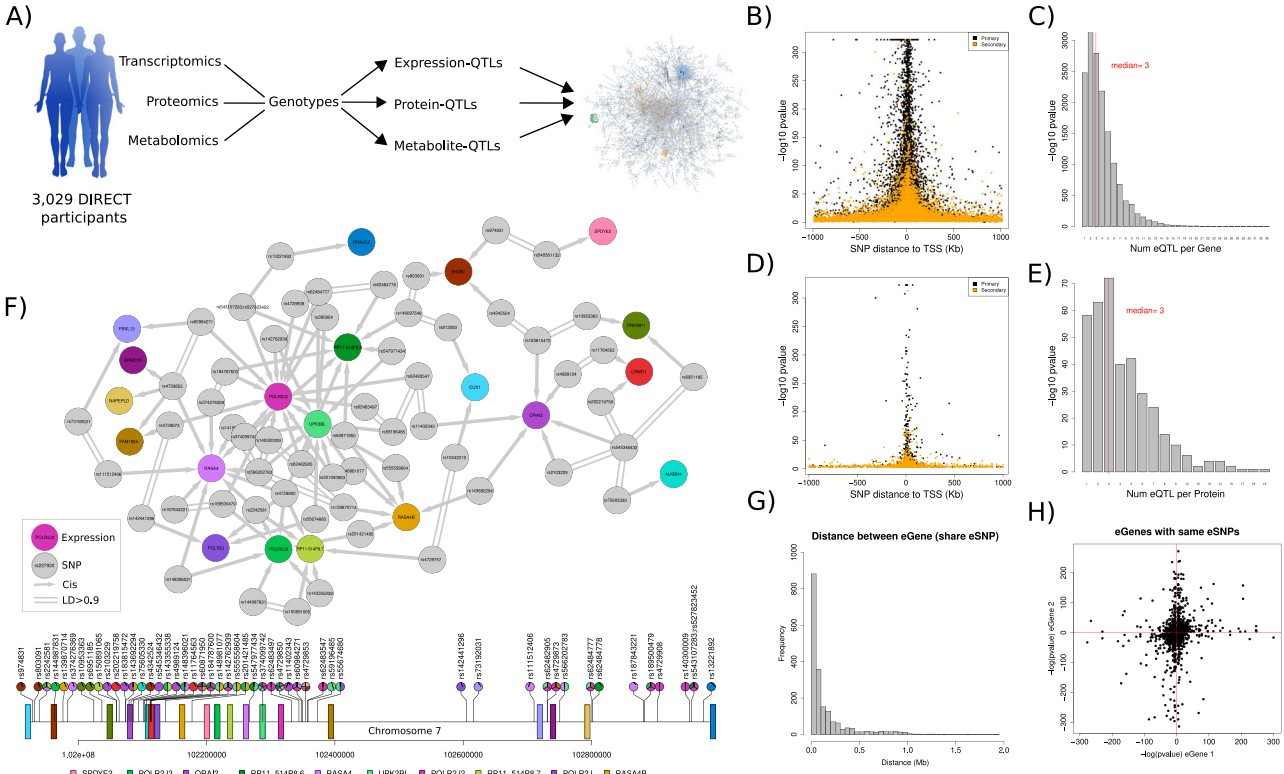

**Fig. 1 | Multiomic QTL analysis identifies extensive allelic heterogeneity and pleotropic effects across molecular phenotypes. A** The DIRECT consortium derived genetic, transcriptomic, proteomic and metabolite data from blood and plasma samples from 3029 individuals. Significant genetic associations (FDR < 0.05) after linear regression between the molecular phenotypes and SNPs in cis (cis-QTLs) and trans (trans-QTLs or GWAS) were used to build a network of genetic perturbations affecting molecular phenotypes. Partially created with BioRender.com. **B** Location of the lead eSNP with respect to the TSS of the significantly associated genes (FDR < 0.05) for cis-eQTL. The most associated eSNP per gene (primary) is shown in black (n = 15,305). Secondary cis-eQTL are shown in orange (n = 44,667). Data shows the -log10 *P* values of the linear regressions between gene expression and SNPs, n = 3029. **C** Number of cis-eQTLs per gene, ranging from 1 to 38. **D** The location of the SNPs acting as cis-pQTLs (pSNPs) centred around the TSS

of the coding gene. The most associated pSNP per protein (primary) is shown in black (n = 373). Secondary cis-pQTL are shown in orange (n = 1217). Stronger cis-pQTLs were significantly closer to the canonical TSS of the gene coding the protein than secondary signals (Wilcoxon test = 9.54e-25). Data shows the -log10 *P* values of the linear regressions between proteins abundance and SNPs, n = 3027. **E** Number of cis-pQTLs per protein, ranging from 1 to 19. **F** Integration of cis-eQTLs identified the largest cis-network of local regulatory genetic effects for genes around *POLR2J2*. The lower lollipop plot shows the genomic location of the genes (boxes) and SNPs (lollipops), coloured by the associated genes. **G** Abundance of genes sharing the lead cis-eSNPs ordered by the distance between the TSS of the pair of genes in Mb. **H** Pairs of genes with the same lead eSNP (n = 583). Data show the -log10 *P* values of the linear regressions between gene expression and the common SNPs, adjusted by the direction of the effect of the eSNP.

Supplementary Fig. 1A–F). For example, 83.8% of the genes and 86.7% of the proteins with cis-QTLs were associated with multiple SNPs, demonstrating extensive allelic heterogeneity. However, functional enrichment analysis of eSNPs relative to pSNPs, using available ChromHMM annotations from 14 blood cell lines[13] and VEP annotations[14], found these two classes of regulatory variants had different properties. pSNPs were enriched in 5' UTR variants relative to eSNPs (OR = 2.84, $P$ value = 8.6e-16) and in variants in zinc finger protein binding sites (up to OR = 10.34, P value = 1e-3), a common DNA-binding motif involved in protein-DNA interactions. On the other hand, eSNPs were enriched in active transcriptional start sites (TSS) (OR E044 = 4.95, $P$ value = 7.7e-03) (Supplementary Fig. 1G and 1H, Supplementary Data 4). In summary, the abundance of independent local genetic regulation identified here supports extensive allelic heterogeneity regulating molecular phenotypes[12].

Pleiotropic effects, when one SNP affects multiple molecular phenotypes, were also common among cis-QTLs and identified networks of local regulatory genetic effects shared across nearby genes (Fig. 1F). For cis-eQTLs, these networks included 1,924 examples of one SNP (eSNP) associated to the expression of two or more genes (Supplementary Data 5). For proteins with limited number of phenotypes ($n$ = 373), just 3 SNPs were associated (pSNP) to two proteins each, all with the same direction of effect: rs2405442 regulated PILRB and PILRA, rs1130371 CCL18 and CCL3, and rs7245416 CD97 and EMR2 (Supplementary Fig. 2A). For cis-eQTLs, genes with shared eSNPs had a mean distance of 0.14 Mb, with 30.30% of the pairs showing the opposite direction of effect for the two cis-eQTLs (Fig. 1G–J). For example, the C allele in rs907612, a variant previously associated with monocyte abundance[15], increased gene expression of the lymphocyte-specific protein 1 (LSP1) while decreasing the gene expression of IFITM10, a gene that codes for the interferon induced transmembrane protein 10 (Supplementary Fig. 2B). To better understand how pleiotropic SNPs may affect gene expression, we annotated genes to topologically associated domains (TADs) called in 8 different blood cell types[16]. Across the different cell types, we found that on average 87% of the pleiotropic eSNPs were only associated with genes within the same TAD, while 13% were associated with genes in 2 different TADs (Supplementary Data 6). Moreover, we observed that pairs of genes associated with a variant with opposite direction of effect were more likely to be further away from each other than those where the variant had the same direction of effect on both genes (Wilcoxon test $P$ value = 7.16e-15), and were more likely to be located on different DNA strands (OR = 1.52, Fisher test $P$ value = 4.21e-06). An enrichment analysis of SNPs associated to two or more genes with opposite direction of effects ($N$ = 583 eSNPs) using VEP found enrichment for "SNPs downstream genes" (Supplementary Fig. 2C). This suggests that the location of the effect allele with respect to the gene body, e.g.: up- or downstream the gene TSS, influences the effect it may have in regulating the expression of the genes. The enrichment analysis using ChromHMM annotations found SNPs with opposite directions of effects were enriched in active enhancers for multiple cell types from peripheral blood (OR = 10.4, $P$ value = 2.38e-03); while SNPs with same direction of effect were enriched in regions classified as "transcription" and "transcription regulation" among others (Supplementary Fig. 2D). These suggest those eSNPs with the same direction of effects on multiple genes may have a more direct and stronger effect on expression by promoting transcription, while those with opposite effects may have effects mediated by other factors such as enhancer regulation. In summary, our results support previous reports of abundant pleiotropic effects by cis-eQTLs[12] with limited information for cis-pQTLs. Given the increased number of proteins and samples evaluated in pQTL studies and reports of overlapping genetic architecture properties with gene expression[17–19], we expect these pleiotropic effects to be also abundant at the protein level.

## Distal genetic regulation shared properties with local regulation

Distal genetic regulation also exhibited allelic heterogeneity and pleiotropic genetic effects for gene expression, proteins and metabolites. To identify trans-QTLs, we first performed genome-wide discovery analyses identifying 1,670 (11.04%) and 139 (37.26%) significantly associated SNPs with gene expression and proteins, excluding a 5 Mb window around the TSS of each phenotype (Methods). Metabolites, which do not have a genomic location of reference, such as a TSS, were evaluated without excluding any window, identifying 172 metabo-QTLs (49.2%). Using a conditional analysis scan we identified independently associated trans-QTLs for each of the three phenotypes. Similar to cis-QTLs, we found evidence that multiple variants were often associated with a phenotype in trans, with an average of 1.38 independent trans associations discovered for each gene, 1.18 for proteins and 1.75 for metabolites (Supplementary Table 1, Supplementary Data 7–9, Supplementary Fig. 3A–D). Similarly, 20.65% of genes with at least one trans-QTL were associated with 2 or more variants, compared to 25.46% of proteins and 39.53% of metabolites. In contrast, pleiotropic regulation was less common for proteins with 8.24% of the pSNPs associated with 2 or more proteins in trans, compared to 14.57% and 17.79% of the SNPs associated with gene expression and metabolites in trans (Supplementary Fig. 3E).

Trans effects on gene expression have been shown to also act as cis-eQTLs[20]. For example, a local effect on the expression of a transcription factor can have downstream consequences on the gene expression of a distal gene. To investigate the common regulatory processes between cis and trans regulation, we looked for trans-QTLs which also acted as cis-QTLs and found significant cis-eQTL effects for 19.39% ($n$ = 262) of 1,351 trans-eSNPs (Supplementary Fig. 4A–H). For proteins, we found no trans-pSNPs that were also acting as cis-pSNPs. Given that these comparisons are limited by differences in significance thresholds and multiple testing, we estimated the proportion of significant trans-eSNPs that also affected local molecular phenotypes and applied the π1 methodology to estimate the proportion of associations from the alternative hypothesis[3,21]. We estimated that 77.34% of trans-eSNPs and 0% of trans-pSNPs had an effect on local gene expression and proteins, respectively (Supplementary Fig. 4I–L). This estimate was higher for trans-pSNPs acting also as cis-eQTLs (91%), while there was no evidence of trans-eSNPs acting as cis-pSNPs (π1 = 0%). Next, we investigated if cis-eSNPs that also have an effect in trans were more likely to regulate a transcription factor (TF)[22] in cis (Methods) and found a significant enrichment of TFs in genes for which the cis-eQTL was also associated with distal genes (OR = 2.26, $P$ value = 5.09e-08). In conclusion, our results support a complex interplay between local and distal regulation of the same phenotypes where trans-QTLs activity involved local regulation of both gene expression and proteins. These distal effects were often driven by local regulation of TFs for trans-eQTLs.

## Phenotype and tissue-specific genetic regulation

We next explored the shared genetics regulation across molecular phenotypes by comparing cis-eQTLs with cis-pQTLs. Of the 373 proteins investigated, 287 had gene expression available for the coding gene in whole blood, while the expression of 86 genes was not detected (Supplementary Data 10). For the available gene-proteins pairs, we compared the $P$ value distribution of significant cis-pQTLs with $P$ values for the same SNP-gene pair from the cis-eQTLs analysis (Fig. 2A–B). We observed a $P$ value enrichment of 66.58% for all pSNPs (73.87% considering only the stronger pQTLs), suggesting a large proportion of cis-eQTLs acted also as pQTLs, even when they were not individually significant. Of the genome-wide significant cis-QTLs detected, 101 cases had a SNP associated with both a protein and the expression of a gene (20.9% of the proteins). For example, rs34097845 was significantly associated with both the gene expression of MPO and its protein (MPO) with a consistent direction of effect (Fig. 2C). Of

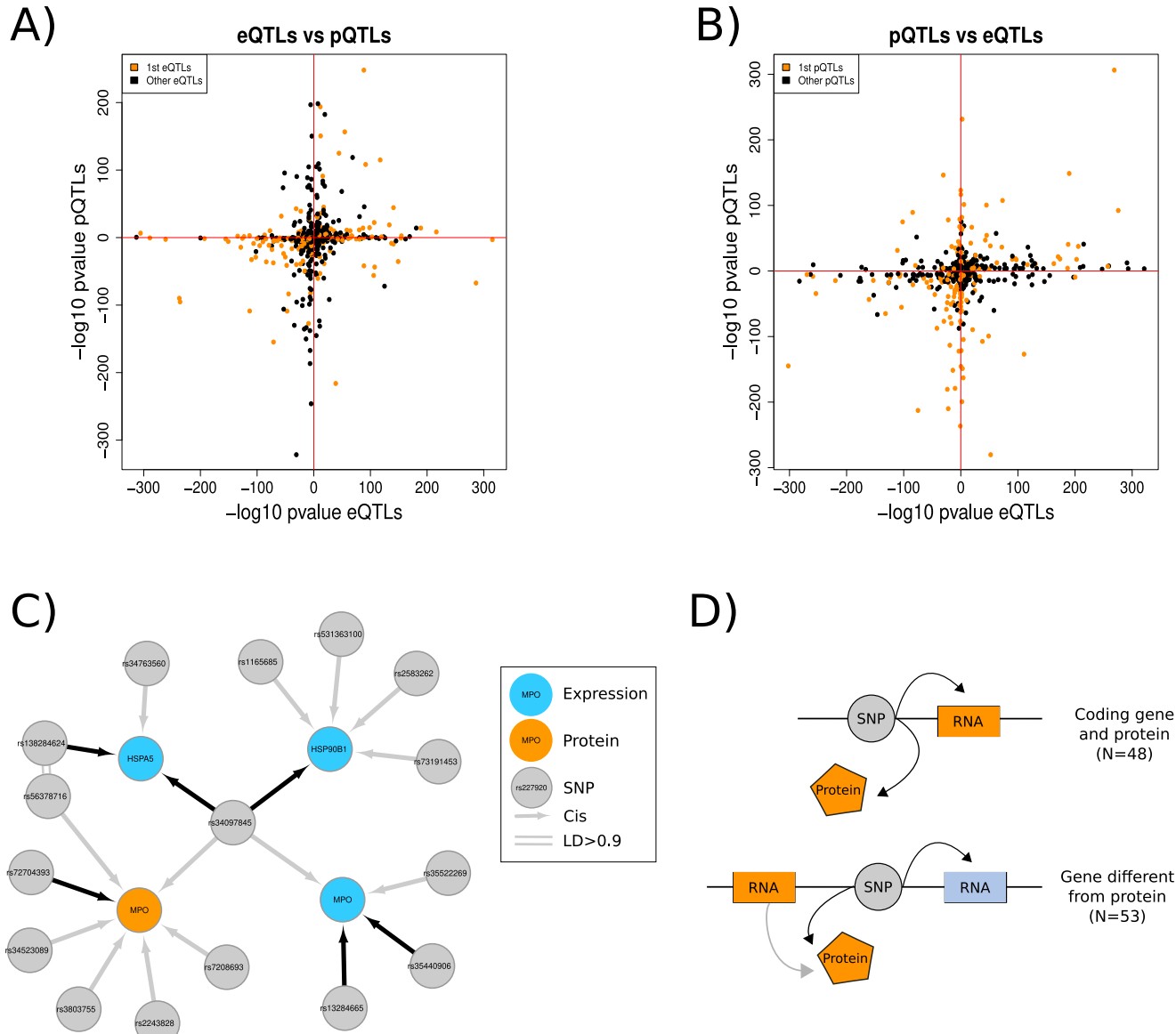

**Fig. 2 | Abundant pleiotropy identified across molecular phenotypes. A, B**
Distribution of the *P* values for SNP in significant (FDR < 0.05) cis-eQTL (**A**) as
pQTLs and for SNPs in significant (FDR < 0.05) cis-pQTLs (**B**) as eQTLs. Most pairs
showed consistent direction of effect. Data shown are the -log10 *P* values of the
linear regressions between gene expression or protein abundances and SNPs.
**C** Local network of QTLs for rs34097845, a SNP significantly associated with both

the expression of *MPO* (*P* value = 1.7e-10, blue) and its protein (MPO, *P* value =
2.08e-14, orange) with a consistent direction of effect ($\beta_{expression} = -0.87$,
$\beta_{protein} = -0.40$). **D** We identified 101 trios of expression-SNP-proteins, of which 48
involved a protein and its coding gene, while 53 involved the expression of a nearby
gene different that the coding gene for the protein.

those, 53 SNPs were associated with a different gene than the coding
gene of the protein. From the 48 SNPs associated to a protein and its
coding gene, 4 had an opposite direction of effect for the cis-QTLs
effect (Fig. 2D, Supplementary Fig. 5A–D). In conclusion, we estimated
a large proportion of shared genetic regulation across gene expression
and proteins. However, as reported by others before[5,17–19,23,24], the
genetic regulation of circulating proteins seems to involve additional
protein-specific regulatory processes that complicates the identifica-
tion of genomic regions acting in both phenotypes.

Failure to identify shared cis-QTLs across molecular phenotypes
may also be driven by tissue-specific genetic regulation. For example,
CCL16 is a cytokine which has a strong, replicated cis-pQTL in whole
blood (lead-pSNP rs10445391, *P* value = 9.57e-245). However, we did
not discover a corresponding cis-eQTL for this gene, as the gene is not
expressed in whole blood. GTEx v8[12] reports the gene to be expressed
mainly in the liver, with rs10445391 acting as a cis-eSNP with the same

direction of effect as the blood cis-pQTL. This suggests that the cis-
pQTL in the blood is the downstream consequence of gene expression
genetic regulation in the liver (Fig. 3A), and demonstrates the need to
put single tissue molecular phenotypes into a whole organism context.

To further investigate the relationship between blood and plasma
molecular associations and similar processes in other tissues, we
looked for evidence that cis-eQTLs detected in other tissues (49 GTEx
v8 tissues[12] and pancreatic islets[3]) were also active in the DIRECT data
from blood. Using *P* value enrichment analysis ($\pi_1$) (Methods), we
compared the distribution of *P* values for significant cis-eQTLs for
genes expressed across different tissues in DIRECT blood eQTLs,
estimating that between 91.2% (pancreatic islets) and 71.6% (esophagus
mucosa) of those cis-eQTLs were also active in whole blood (Fig. 3B).
Our results indicate that a sufficiently large sample size in blood can be
informative of the regulation of genes expressed in other tissues,
although a specific genetic process acting on specific genes, such as

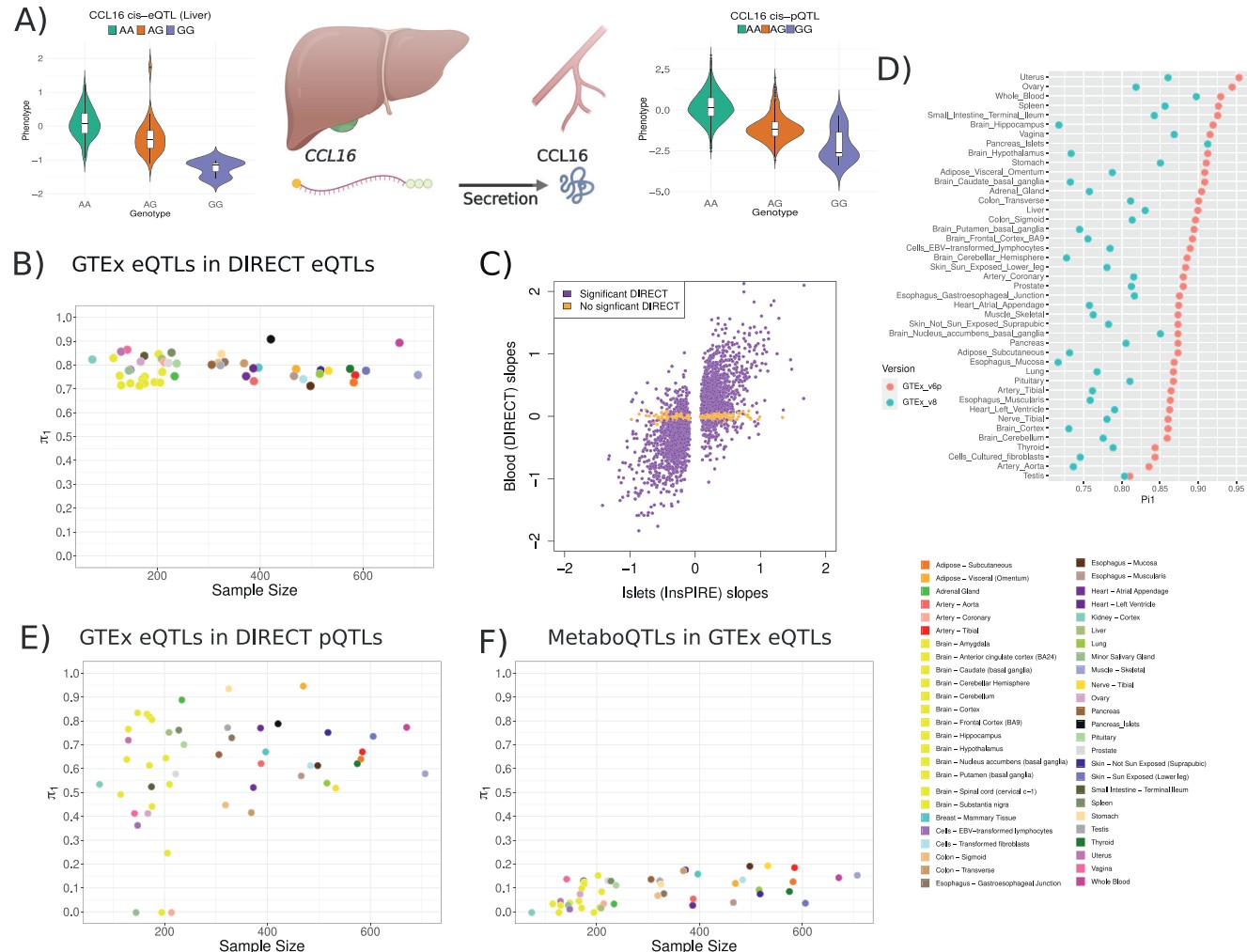

**Fig. 3 | Tissue specific genetic regulation partially explains the lack of shared associations between gene expression and proteins. A** Using $n = 3027$ biologically independent samples, we detected a cis-pQTL for CCL16 in whole blood ($P$ value = 9.5e-243, $n = 3029$). The GTEx consortium reported a cis-eQTL, with the same SNP (rs10445391) affecting the expression of the gene in liver ($n = 208$). Violin plots show the median and first and last quartiles as defined by ggplot geom_violin function. Partially created with BioRender.com **B** Between 91.2% (pancreatic islets) and 71.6% (esophagus mucosa) of cis-eQTLs discovered by GTEx v8 were also active in whole blood DIRECT datasets ($n = 3029$) as shown by the $\pi_1$ values (y-axis). The number of $P$ values per tissue used to calculate the $\pi_1$ estimates ranged from 334 in kidney to 14,920 in thyroid. **C** Comparison of the effect size of cis-eQTLs from pancreatic islets (InsPIRE) and whole blood (DIRECT). A total of 486 eQTLs were not significant in blood ($P$ value > 0.035, orange color) but significant in pancreatic islets ($n = 420$) and 294 had opposite direction of effect ($N = 2691$). Data shown are

the ß values (effect) resulting from the linear regressions between gene expression and SNPs identifying eQTLs in both studies. **D** Comparison of the $\pi_1$ enrichment analysis between an earlier version of GTEx (v6p) and a larger later version (v8). eQTLs from DIRECT blood detected in GTEx v8 decreased compared to v6p independently of the change in sample size across versions (Supplementary Fig. 5H). **E** Degree of sharing of pQTLs detected as eQTLs in GTEx v8 tissues. Up to 66.6% of plasma cis-pQTLs were also active as DIRECT whole blood cis-eQTLs. The number of overlapping QTLs across tissues oscillates between 13 (kidney) and 311 (Thyroid). **F** Degree of sharing of metabo-QTLs acting as cis-eQTLs in GTEx v8. Up to 16.88% (testis) of the metabo-QTLs detected in blood were active eQTLs in other tissues, with many tissues sharing no associations with metabolites-QTLs. The number of $P$ values used to calculate $\pi_1$ values per tissues ranged from 4298 in whole blood to 6575 in testis.

the effect of rs7903146 on *TCF7L2* in pancreatic islets, may be missed if relevant tissues are not studied (Fig. 3C, Supplementary Fig. 5E–G). However, a comparison of these estimates with an earlier version of GTEx with a smaller sample size per tissue (v6p[25]), showed the increase in sample size for cis-eQTLs studies in less accessible tissues reduced the degree of shared genetic effects detected across tissues and phenotypes (Fig. 3D, Supplementary Fig. 5H). Next, we investigated whether these cis-eSNPs active in other tissues were also regulating protein or metabolite levels from blood plasma. For cis-pQTLs, we observed π1 estimates ranging from 0% (artery coronary tissue or minor salivary gland) to 94.9% (adipose visceral omentum, Fig. 3E). However, these estimates were calculated using only between 13 and 311 $P$ values, as the number of pQTLs was limited. These estimates therefore showed large confidence intervals (Supplementary Fig. 5I–K), indicating a

limited value to evaluate the level of activity of cis-eQTLs from multiple tissues acting as cis-pQTLs in plasma. For metabolites with no direct match between phenotypes, we extracted all the cis-eQTLs or cis-pQTLs associated with the most significant metabo-SNPs to calculate π1 estimates. Metabo-SNPs acting also as cis-eQTLs and cis-pQTLs in DIRECT whole blood were of 33.63% and 27.03%, respectively. These estimates are lower than the cis-pQTLs detected as cis-eQTLs (66.6%) and suggest a higher degree of independent regulation across those phenotypes. Enrichment estimates for metabo-SNPs active in the 49 GTEx tissues were between 0% and 19.6% (Fig. 3F), suggesting a limited direct relationship between the genetic regulation of these phenotypes across tissues. Overall, our results indicate that blood gene expression and protein levels share a large degree of genetic regulation, both within and across tissues, and to a much greater degree than

those shared with metabolites. Likewise, our comparison between studies with different sample sizes highlights the value of increasing the number of samples for the study of genetic regulation of molecular phenotypes, as more tissue and molecular-specific regulation will be identified.

Next we assessed the ability of the DIRECT data set to identify distal genetic associations in other studies and tissues. Using GTEx v8, we tested an average of 1662 gene-SNPs trans-eQTLs pairs per tissue (range = 1,512-1,760) and found that blood trans-eQTLs were also observed in whole blood ($\pi 1 = 0.336$) and brain putamen basal ganglia ($\pi 1 = 0.136$) among others, while 15 tissues did not identify any significant replicated trans-eQTLs from DIRECT blood ($\pi 1 = 0$, Supplementary Data 11). After multiple testing correction, we replicated 278 significant blood trans-eQTLs (237 unique genes, 159 unique SNPs), corresponding to 643 gene-SNP pairs across all tissues. In contrast, only 4 trans-pQTL were observed as trans-eQTLs in GTEx tissues after multiple testing correction: FCRL5-rs569841457 (adipose subcutaneous), MMP9-rs919377 adrenal gland, SPINK5-rs12462111 (nerve tibial) and OLR1-rs76604815 (uterus). Additionally, we evaluated the number of significant blood trans-eQTLs, trans-pQTLs and metaboQTLs that replicated in other blood and plasma datasets. Using the eQTLGen dataset[26], we were able to evaluate 514 gene-SNP pairs from DIRECT trans-eQTLs, of which 463 were also significant (Supplementary Data 11). For cis and trans-pQTLs replication we used GWAS summary statistics from Sun et al. [17] and found that 281 cis-pQTLs and 65 trans-pQTLs affecting 253 proteins replicated. For metabolites, we were able to evaluate 65 metabolite-SNPs pairs from 47 metabolites, of which all of them replicated in Long et al. [27] (Supplementary Data 11).

## Causal networks

Given the abundant pleiotropy observed, with single SNPs associated with multiple molecular phenotypes, we were interested in characterizing the chain of action of genetic variation on molecular phenotypes. Therefore, we tested causal networks from 65,682 trios consisting of 14,288 SNPs significantly associated with two molecular phenotypes in cis or trans (Supplementary Fig. 6). These trios are partially directed, as causation must travel outwards from the DNA (QTLs)[5,28]. We used Bayesian Networks (BN, Methods) to evaluate two main types of causative models: i) independent models, where the SNP independently regulates the two phenotypes; ii) dependent models, where the SNP regulation of one phenotype depends on the activity of the other phenotype (Fig. 4A–B). After evaluation of the best models, we identified 23,883 trios of SNPs (or two SNPs in high LD (R2 > 0.9)) with evidence supporting a particular casual model (Supplementary Data 12). All combinations of QTLs and the causative models investigated are fully described in Supplementary Figs. 7 and 8.

The shared genetic regulation between different phenotypes, such as mRNA vs. proteins, had different causative relationships than the shared effects between pairs of the same molecular phenotypes, such as cis-eQTLs vs. cis-eQTLs or cis-eQTLs vs. trans-eQTLs effects (Fig. 4C–D). Dependent models, where the SNP's effect on one phenotype was mediated by the abundance of the other phenotype, were more often supported for trios where the SNP regulates two different molecular phenotypes (e.g.: trans-eQTL/trans-pQTL, cis-eQTL/metaboQTL) than the same molecular phenotype. For example, we found no evidence of SNPs independently affecting cis-gene expression and metabolites, cis-gene expression and trans-protein levels, or trans-gene expression and trans-protein levels. Only 24 out of 303 models for shared effects on cis-eQTLs and cis-pQTLs found the independent model as most likely causal, suggesting the genetic effects of one SNP on both gene expression and proteins is commonly mediated by one of the molecular traits (Fig. 4E–F). An exception of this trend was observed for shared genetic regulation between two metabolites, as it was more often identified as independent (Supplementary Fig. 7D). For trios involving local and distal genetics effects, we found little support

for independent SNP effects. More models were supported with local associations mediating the effects on the distal phenotype when the same type of phenotype was considered (mRNA-to-mRNA), while for different phenotypes more models supported a regulation of the local phenotype mediated by the distal phenotype (Supplementary Fig. 8). In conclusion, our causal network analysis supports a model where the downstream consequences of genetic variation are often mediated by other molecular phenotypes. Counterintuitively, we also found that the path of local-distal regulation can often be mediated by distal regulation acting locally. This suggests that even well-powered studies such as this one are not yet large enough to identify the full extent of regulatory elements involved in the distal regulation of molecular phenotypes.

## Integration of molecular QTLs identifies networks of GWAS variant effects

QTL studies are often used to identify candidate molecules mediating the activity of GWAS variants. However, the huge polygenicity, extensive allelic heterogeneity and pleiotropy reported for even single molecular phenotypes[29] limits our ability to identify candidate gene products, as variant's effects can span multiple genes, assays and tissues. To simultaneously evaluate the regulatory elements of molecular phenotypes and GWAS variants, we constructed networks with nodes representing either genetic variants or molecular phenotypes and edges representing significant associations. This approach allowed us to observe allelic heterogeneity as multiple edges from genetic variant nodes pointing out to a particular molecular phenotype, and pleiotropy as multiple edges from a single genetic variant pointing out to multiple molecular phenotypes. In addition, this data representation moves beyond pairs of QTLs by adding information from GWAS studies, partially visualizing the network of GWAS variant effects, and their effects on mediating molecular phenotypes (Fig. 1A, Supplementary Fig. 9).

The complete network of all connected genetic effects detected on genes, proteins and metabolites included 79,733 nodes (15,254 genes, 373 proteins, 172 metabolites and 63,795 SNPs) and 80,645 edges identifying significant QTLs, connected in clusters containing between 3 and 19,711 nodes. Nodes had an average of 4.31 edges connecting them to neighbouring nodes. To investigate how molecular phenotypes could have downstream consequences for the risk of disease, we extracted information from the GWAS catalogue (GWAS catalogue v1.0.2, accessed 26/10/2020[30]) and identified all SNPs that were lead GWAS variants and acted as QTLs in blood. In our network, we observed 2828 GWAS variants (Table 1, Supplementary Data 13–14) connected with an average to 1.9 molecular phenotypes: in total 823 genes, 58 proteins and 44 metabolites were connected to GWAS variants. Among the traits more often observed to be associated to SNPs in the network, we found blood cell counts (33.01% of the 2,828 variants tested) and plasma metabolites or proteins levels (9.11%), suggesting blood-related traits were better characterized by the network. We also investigated if blood-related traits were more likely to co-localize with cis-eQTLs in blood than other phenotypes commonly studied using GWAS, using data from 16 different studies (Supplementary Data 15) that included blood-related traits such as lymphocyte and platelet counts[15], and other traits such as height[31] and schizophrenia[32](Methods). We found that 72.13% of all blood cell counts variants and 70.69% of all lipid traits variants co-localized with cis-eQTLs (COLOC probability>0.9, Supplementary Data 16, Supplementary Fig. A10). However, we also identified a large number of co-localizing signals with other traits without a clear relationship with blood. For example, 77 of 97 variants (79.38%) associated to height[31] had evidence of co-localization with eQTLs in blood, while we found 26 of 35 for Type 2 Diabetes[33]. All in all, we identified thousands of candidate molecular phenotypes associated to GWAS traits in an accessible tissue, further analysis is required to investigate their potential

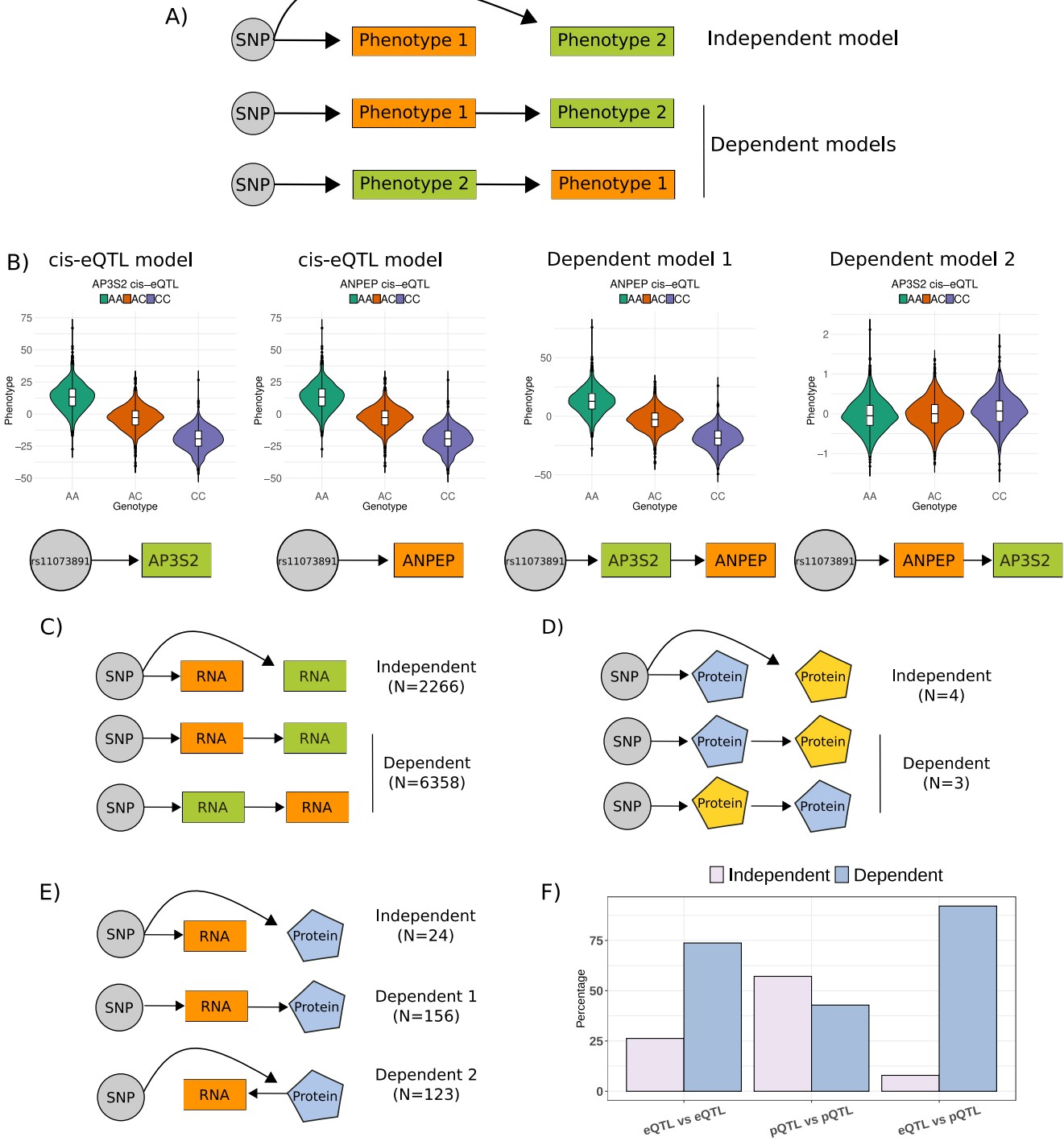

**Fig. 4 | Causal inference identifies distinct patterns of casual paths in the regulation of molecular phenotypes. A** Two main models were tested for casual inference. The dependent model assumes the effect of a genetic variant (SNP) on one phenotype (1 or 2) is mediated by the other phenotype. The independent model assumes the effect of the SNP on both SNPs is independent, and no mediation between phenotypes occurs. **B** Example of model testing for rs11073891 association with the gene expression of *AP3S2* and the expression of *ANPEP* (n = 3029). The results for the dependent model 2, testing for the mediation of *ANPEP* in the SNP effect on *AP3S2* shows a change in directionality consistent with a mediation. Data shown are residuals of expression removing effects of any other eSNP grouped by genotypes ($n_{AA}$=1070, $n_{AC}$ = 1419 and $n_{CC}$ = 507 for all figures).

**C** Casual models testing paths for SNPs acting as cis-eQTLs for two genes identified slightly more models supporting independent effects of the shared eSNPs than dependent effects. The test used n = 3027 biologically independent samples with gene expression. **D** Casual models testing paths for SNPs acting as cis-pQTLs for two proteins identified similar numbers of dependent and independent cases, but only 7 models were conclusive. **E** Casual models for shared SNPs associations between gene expression and proteins supported more often dependent models, with similar proportions where expression was the mediating factor as where the mediating factor was the protein levels. **F** The casual network analysis supports a model where the downstream consequences of genetic variation were often mediated by other molecular phenotypes.

**Table 1 | Summary of GWAS SNPs overlapping with significant QTLs**

| Trait | SNPs | Percentage total |
|---|---|---|
| Total blood cell counts | 1290 | 33.01 |
| Metabolites/proteins | 356 | 9.11 |
| Blood pressure/CAD | 338 | 8.65 |
| Blood protein levels | 248 | 8.77 |
| BMI/obesity | 171 | 4.38 |
| Lipids | 133 | 3.4 |
| Autoimmune/inflamatory | 129 | 3.3 |
| Education/brain/behaviour | 120 | 3.07 |
| Height | 106 | 2.71 |
| **ALL OTHER TRAITS** | **1513** | **38.62** |
| Core binding factor acute myeloid leukemia | 59 | 1.51 |
| Heel bone mineral density | 57 | 1.46 |
| Type 2 diabetes | 44 | 1.13 |
| Multiple sclerosis | 37 | 0.95 |
| C-reactive protein levels | 27 | 0.69 |

We identified all SNPs that were lead GWAS variants and acted as QTLs in blood, identifying 2,828 GWAS variants acting as QTLs. Most SNPs were associated to blood related traits, such as blood cell counts or plasma proteins and metabolites, which are often included as GWAS studies. Full list and summary of GWAS SNPs counts can be found in Supplementary Data files 13 and 14.

causal role in disease given that QTL signals are often shared across tissues[3,12].

Next, we evaluated the type of molecular phenotypes that may be involved in mediating the activity of GWAS variants. Using all SNPs reported as GWAS variants in the network, we found that these were connected to more molecular phenotypes than other variants in the network (P value = 6.7e-97, Wilcoxon test), and were more likely to be associated to proteins (OR = 8.93, P value = 3.15e-25) or metabolites (OR = 17.51, P value = 1e-09) than to gene expression (Supplementary Data 13). Per type of QTL, we observed that GWAS were more often cis- and trans-eQTLs (Fig. 5A). However, when considering the number of tested phenotypes, we saw that a large percentage of SNPs involved in metaboQTLs were also GWAS variants (63.12%), followed by SNPs acting as trans-eQTLs (34.48%) (Fig. 5A). This enrichment could be due to GWAS variants more likely acting via processes not captured by gene expression, such as post-transcriptional modification, or it could be due to lack of statistical power. As we have more power to identify eQTLs, a higher proportion of our gene expression associations represent weak biological effects, with smaller downstream consequences on GWAS traits than proteins or metabolites. To evaluate the influence of statistical power for different phenotypes, we repeated the enrichment considering only the most significant gene expression associations, matching the number of protein associated SNPs or the number of metabolite associated SNPs (Methods). While the relative metabolite enrichment remained significant (OR = 2.99, P value = 1.51e-3), the protein enrichment was reversed (OR = 10.9, P value = 3.28e-81 for gene expression over proteins), suggesting GWAS enrichment was here not driven by post transcriptional processes. Moreover, genetic associations with metabolites and proteins are generally reported as GWAS studies, and included in the GWAS catalogue, driving to some extent the large overlap between both. This is not true for trans-eQTLs signals, for which we observed a large percentage of trans-eSNPs also as GWAS variants for nonmolecular traits and diseases. Overall, we observed that GWAS variants modulated the levels of more molecular phenotypes than non-GWAS variants associated to molecular phenotypes; in particular they were enriched in associations with metabolites and strong genetic effects on local and distal gene expression.

Finally, we highlight here three examples to illustrate the complexities of GWAS variant interpretation and the benefits of understanding the full regulatory context to infer their underlying mechanism of action. First, we evaluated the largest network cluster, with 19,711 nodes (3362 genes, 147 proteins, 15 metabolites and 15,334 SNPs) (Fig. 1A, Supplementary Fig. 9). The cluster was enriched in genes and proteins involved in immune response and hematopoietic cell lineage regulation (Supplementary Data 17) and included 928 SNPs previously associated with multiple GWAS traits/diseases for blood protein levels, platelet counts, and triglycerides. Among the 614 trans-eSNPs included in this cluster, we found a replicated trans-eQTL for rs1354034[26]; this variant was associated with the expression of 297 genes and is involved in platelet function regulation[34,35] Supplementary Fig. 10B). Within this cluster, we also found the resistin gene (RETN), previously associated with low-density lipoproteins (LDL) levels and cardiovascular disease[36] (Fig. 5B). RETN expression was associated with two trans-eSNPs: rs13284665, also associated with the expression of 16 other genes, and rs149007767, previously associated with blood cell counts[15] and responsible for the regulation of 67 other genes in our study. This second trans-eSNP also acted on two genes in cis, the growth factor receptor-bound protein 10 (GRB10) and the transcription factor IKAROS family zinc finger 1 (IKZF1) with opposite direction of effect, suggesting both would be mediating candidates genes for the trans effect on RETN (Supplementary Fig. 10C). Overall, this example highlights the shared genetic regulation between hematopoietic production and lipid metabolism, and demonstrates the complex network of multiple molecular phenotypes mediating GWAS variants activity.

In a second example we focus on the cis-window around the FADS1 gene on chromosome 11 (Fig. 5C). This region contains a cis-eSNP (rs968567) associated with FEN1, FADS1 and FADS2 expression which has previously been associated with rheumatoid arthritis[37] and fatty acid desaturase activity[38]. The region also harbours a complex locus with multiple replicated GWAS associations for metabolites[4,39]. Our results suggest multiple independent SNPs regulate fatty acid related metabolites, supporting the mediation of FADS1/FADS2 in regulating plasma metabolites levels such as arachidonate (20:4n6)[40], while also identifying other candidate effector transcripts mediating metabolites regulation such as TMEM229B or FEN1. The third example involves a shared genetic regulation of the interleukin-6 (IL-6) and its receptor (IL-6R) (Fig. 5D). Both expression and protein levels of IL-6 were locally regulated by rs11766947 in addition to other cis-SNPs on chromosome 7. IL-6 proteins levels were also associated with a trans-pSNP on chromosome 1 in high LD ($R^2$ = 0.97) with a replicated cis-eSNP (rs12133641) for the IL-6R gene (also known as IL-6-RA). These networks support the hypothesis that shared genetic regulation may modulate IL-6 levels, which increase after treatment with anti-IL-6 receptor antibodies for rheumatoid arthritis[18].

## Discussion

Using genetic association analyses, we have evaluated the interplay between blood mRNA molecules, proteins and metabolites, as well as their genetic basis. We used genetic perturbations which allow for the orientation of effects and causal modelling to identify the regulatory principles of molecular phenotypes and potentially the paths of genetics-to-disease in humans. We report large and complex cis-regulatory networks connected across different regions of the genome by trans-effects. Local allelic heterogeneity and pleiotropic effects have been reported for expression and proteins[5,12,17,18,41–43], but we observed here cis and trans allelic heterogeneity at larger scales than previously reported. This supports the presence of additional downstream regulation for proteins independent of gene expression regulation and provides an explanation for the lack of correlation between expression and protein levels reported previously[23].

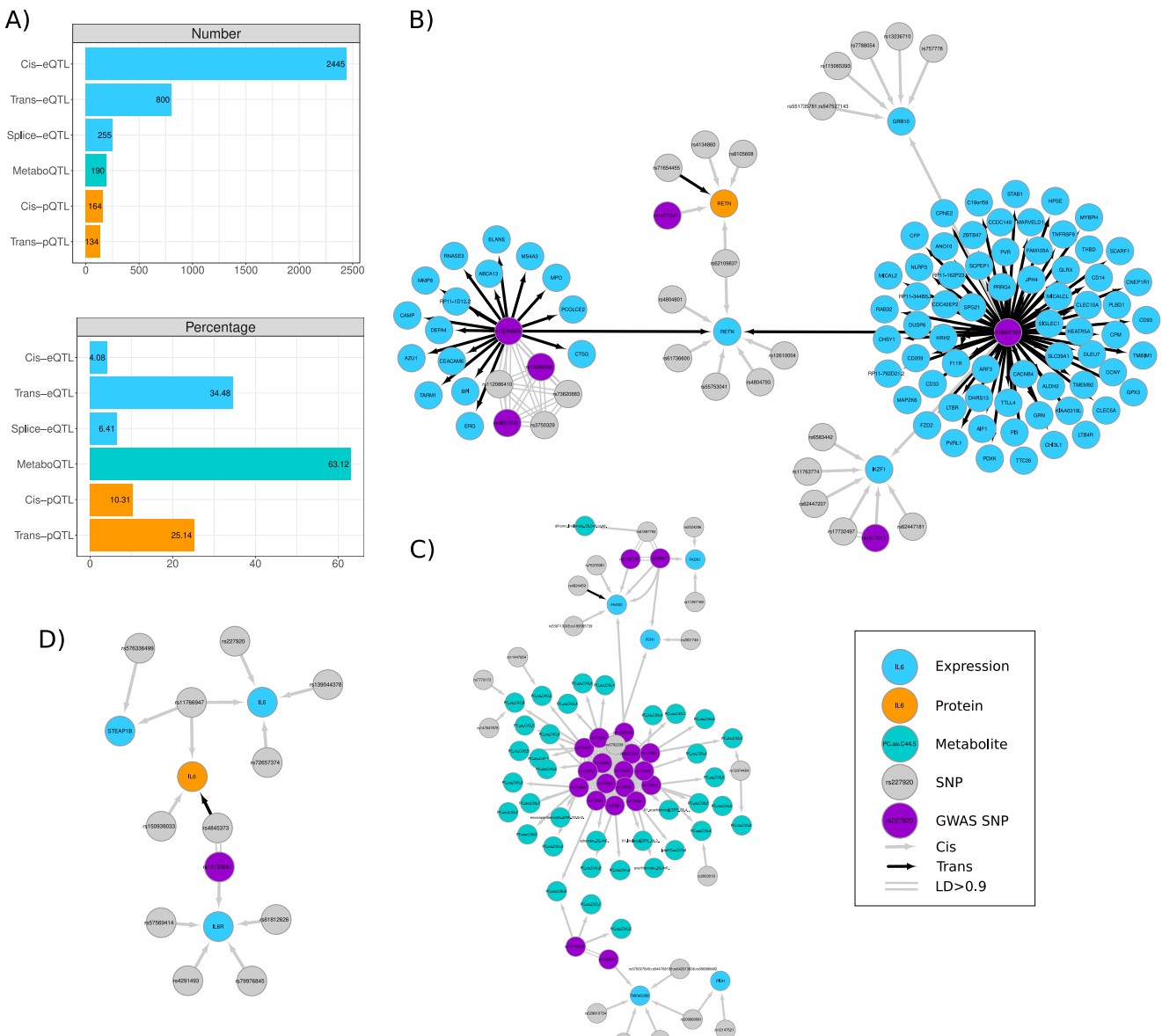

**Fig. 5 | QTL integration identifies regulatory networks associated to GWAS variants. A** Of the GWAS signal overlapping SNPs in the full network (Supplementary Fig. 9), the largest number were cis-eSNPs followed by trans-eSNPs (Number). However, when considering the number of significant QTLs evaluated (Percentage), we observed that more metabo-SNPs were also reported GWAS followed by trans-eSNPs. The barplots show numbers and percentages of SNPs involved in QTLs that were also reported as lead GWAS by the GWAS catalogue (Supplementary Data 13). **B** Network of associations for the resistin gene (*RETN*). The *RETN* gene and its protein (orange node) have been associated with low density lipoproteins (LDL) levels. The regulatory network associated with the gene included GWAS variants (purple nodes) associated to RETN abundance (rs1477341); cardiovascular diseases and cholesterol levels (rs13284665); platelet counts

(rs13284665, rs13284665, rs149007767) and monocyte counts (rs149007767). **C** Network for the *FADS1/FADS2* genes centred in a cis-eSNP (rs968567, purple) associated with *FEN1*, *FADS1* and *FADS2* and reported as lead GWAS association for lipid metabolism. The network shows their relationship with a cluster of genetic associations with metabolites (metabo-QTLs), many of which have been reported by other studies. **D** Network for the Interleukin-6 (*IL6*) gene. This network shows an example of a SNP in chromosome 7 (rs11766947) acting as cis-eQTL and cis-pQTL for both the gene expression and the protein levels for the same gene, *IL6*. The network shows a shared genetic regulatory effect for IL6 and the expression of its receptor *IL6R* mediated by a trans-eQTLs signal (rs4845373, chromosome 1) for IL6 and in high LD (R2 > 0.9) with rs12133641, a splice-QTL for the IL6-receptor (*IL6R*).

We evaluated the extent to which molecular genetic analyses on whole blood and plasma can inform us about molecular genetic regulation in other tissues. Early multi-tissues eQTL studies found a large proportion of genetic effects shared across tissues[9,25]. With the increase in the number of samples and the genetic diversity included, similar studies report now a lower degree of sharing across tissues[12], with more cis-eQTLs with weaker effects and tissue specific effects. However, when we attempted to identify cis-eQTLs from other tissues, we could detect between 71.6% and 91.2% of these signals in our whole blood dataset as active in genetic regulation in other tissues.

This suggests that a sufficiently large sample size in an accessible tissue can be informative of regulation in other tissues. However, we also reported up to a 20% decrease in the degree of sharing genetic signal across tissues when the sample size of the GTEx dataset increased with the transition from version 6 to version 8. Since this decrease was not uniform across tissues, researchers should evaluate carefully what accessible tissue may be most suitable for their research.

The choice of tissue for the study of genetic regulation is even more critical for those aiming to understand the underlying

mechanism of GWAS variant effects on molecular phenotypes. A clear example is the *TCF7L2* loci associated with T2D. Despite finding 6 independent cis-eQTLs for the gene, none of these involved the T2D loci, which has only been shown to have an effect on expression in pancreatic islets[3]. Therefore, the tissue of choice to study complex traits remains critical for research involving the activity of specific variants. This is confounded with the difficulty in defining the relevant tissue for a given disease or trait[2,44]. First, such definitions imply complex diseases such as T2D or cardiovascular diseases are driven by genetic and environmental factors active only in one or a handful of tissues. However, we have been able to identify a large number of co-localizing QTLs with GWAS signals from traits without a clear relationship with blood. One reason, particularly when working with circulating proteins and metabolites, is that those molecules may not have been produced in blood, reflecting genetic effects in non-accesible tissues, e.g.: *CCL16*. Other reasons include a large degree of sharing of genetic regulation across tissues, cross tissue regulation, and cross phenotype regulation. Until we learn the general principles of regulation of multiple molecular phenotypes and the possible coordinated impact of genetic variation in and across tissues, our knowledge about how genetic effects cascade from the specific alleles to gene expression and other molecules to drive disease will remain limited. This is only currently possible using accessible tissues.

Finally, our work provides a roadmap to understanding the underlying mechanisms of action of GWAS variants. Using networks derived from thousands of QTLs, we observed the intricate connections across molecular components regulated throughout the whole genome, rather than a single direct pathway from genotypes to phenotypes. Our results support models of genetic regulation that consider thousands of small and coordinated genetic regulatory effects across the genome to modulate complex traits[29,45–47]. Moreover, our results suggest that the regulatory process that connects genotype to phenotypes is robust, with redundancies in the form of many connections between molecular phenotypes and the ability to find alternative routes in the event of a particular process being altered by disease-related variants. Therefore, as proposed by others[48], we have observed how a network of variants and its connected molecules as a whole is likely required to define a given phenotype or disease.

## Methods

### Cohort

The DIRECT (Diabetes Research on Patient Stratification) consortium (Supplementary Data 19) includes pre-diabetic participants (target sample size 2,200–2,700) and patients with newly diagnosed type 2 diabetes (target sample size -1,000) with detailed metabolic phenotyping. Characteristic of the cohort as well as inclusion/exclusion criteria have been described elsewhere[49]. In short, fasting blood samples from venous blood were collected and DNA extractions and other biochemical analyses were carried out as detailed bellow. All measurements for each molecular data type were taken from different samples. Ethics approval for the study protocol was obtained by all the regional research ethics review boards (Lund, Sweden: 20130312105459927; Copenhagen, Denmark: H-1-2012-166 and H-1-2012-100; Amsterdam, Netherlands: NL40099.029.12; Newcastle, Dundee, and Exeter, UK: 12/NE/0132). Participants gave informed consent at enrolment in writing. The research conformed to the ethical principles for medical research involving human participants outlined in the declaration of Helsinki. The cohort included 2142 men and 887 women with a mean age of 61.6 years old (yo), and a range of 30yo to 75yo. Sex was determined by matching genotype information and self reporting information. Analyses included sex as a co-variate.

### Genotyping

Genotyping was conducted using the Illumina HumanCore array (HCE24 v1.0) and genotypes were called using Illumina's GenCall algorithm. Samples were excluded for any of the following reasons: call rate <97%; low or excess mean heterozygosity; gender discordance; and monozygosity. Genotyping quality control was then performed to provide high-quality genotype data for downstream analyses using the following criteria: call rate <99%; deviation from Hardy-Weinberg equilibrium (exact p < 0.001); variants not mapped to human genome build GRCh37; and variants with duplicate chromosome positions. To identify possible ethnic outliers in the DIRECT data, we performed a principal component analysis (PCA) using the genotype data from our studied population (3102 samples; 547,644 markers) using the following cut-offs MAF > 0.01, HWE>1e-4 and call rate>90%. A total of 3033 samples and 517,958 markers across the two studies passed quality control procedures. Imputation to the 1000 Genomes Phase 3 CEU reference panel was performed with ShapeIt (v2.r790)[50] and Impute2 (v2.3.2)[51].

### RNAseq data generation

mRNA samples were processed and quality was assessed using the TapeStation Software (A.01.04) with an RNA Screen Tape from Agilent. Truseq Stranded mRNA from Illumina was used to generate libraries and their quality was evaluated using Qubit and TapeStation using DNA1000 Screen Tape. The samples were then sequenced on the Illumina HiSeq2000 platform using 49 bp paired-end reads. The 49-bp sequenced paired-end reads were mapped to the GRCh37 reference genome[52] with GEMTools 1.7.1[53]. Exon quantifications were calculated for all elements annotated on GENCODE v19[54]. Gene quantifications were calculated as FPKM values. This pipeline is fully described in Delaneau et al. [55], as part of QTLtools. Samples with a total number of exonic reads lower than 5e + 06 reads or with a proportion of exonic reads over the total number of reads lower than 20% were considered of low quality and removed. For each samples, we evaluated possible samples mix-ups[56] using the function *match* from the suite QTLtools[57]. To confirm the correct assignment of the matched DNA/RNA samples and recovered failed genotypes during QC we re-genotyped samples from 96 individuals. Further validation compared the sex information provided by clinical reports with both genotype data and RNAseq data. The total number of samples with RNAseq-Genotypes pairing data after QC of both RNAseq and imputed genotypes was 3029.

### Expression phenotypes

Genes and exons with more than 50% of zero reads were removed from the study. Finally, exons and genes from chromosome Y, mitochondria, and level 3 annotations, as defined by Gencode v19, were removed from further analysis. The final number of genes and exons used for analyses were 16,205 and 170,198, respectively. Splicing phenotypes were generated using LeafCutter[58] requiring a minimum of 50 reads per cluster. Clusters with more than 50% zero reads across samples were removed. The final number of phenotypes used for further analyses was 64,546.

### Proteins data

Plasma proteins were measured using the Olink® Cardiometabolic, Cardiovascular II, Cardiovascular III, Development and Metabolism panels (Olink Proteomics AB, Uppsala, Sweden) according to the manufacturer's instructions. The Proximity Extension Assay (PEA) technology used for the Olink protocol has been well described[59]. The obtained data was processed using Olink's NPX manager software version 0.0.85.0. Internal and external controls were used for quality control and normalization of the data. Quality control included calculating the standard deviation for the detection control and the incubation/immuno controls and comparison of the results for the

detection control and one of the incubation controls against the run median. All protein measurements were reported, but proteins with more than 50% samples below LOD levels were excluded, leaving a total of 373 proteins for further analysis from 3027 individuals.

## Metabolites data

Metabolites abundance was assessed using targeted and untargeted technologies. The assays, quality control and analyses were performed separately, and results combined for discussion. Plasma targeted metabolites were evaluated for 163 metabolites using a FIA-ESI-MS/MS-based targeted metabolomics approach with the Absolute*IDQ*™ p150 kit (BIOCRATES Life Sciences AG, Innsbruck, Austria) as described in Biocrates manual AS-P150. Mass spectrometric analyses were done on an API 4000 triple quadrupole system (Sciex Deutschland GmbH, Darmstadt, Germany) equipped with a 1200 Series HPLC (Agilent Technologies Deutschland GmbH, Böblingen, Germany) and a HTC PAL auto sampler (CTC Analytics, Zwingen, Switzerland) controlled by the software Analyst 1.6.2. Data evaluation for quantification of metabolite concentrations and quality assessment was performed with the software MultiQuant 3.0.1 (Sciex) and the Met*IDQ*™ software package, which is an integral part of the Absolute*IDQ*™ Kit. Five aliquots of a pooled reference plasma were analyzed on each kit plate and used for normalization purposes and for calculation of coefficient of variance (CV) for each metabolite. Quality assessment evaluated peak shapes, retention times, compound identity, and the number of samples with zero values in the metabolites concentration, removing any individual with more than 50% of zeros. We then evaluated the CV per metabolite removing samples with CV > 0.25 relative to the reference samples. Metabolites with concentration below the LOD were discarded. Of the 163 metabolites, 116 passed all quality controls in 3029 individuals.

Untargeted metabolites from human plasma samples were assessed using the Metabolon platform. Controls included a pooled matrix sample generated by taking a small volume of each experimental sample serving as a technical replicate throughout the data set. Experimental samples were randomized across the platform run with QC samples spaced evenly among the injections. Metabolite concentration was assessed using Liquid Chromatography-Tandem Mass Spectrometry (LC-MS/MS). The LC-MS portion of the platform was based on a Waters ACQUITY ultra-performance liquid chromatography (UPLC) and a Thermo-Finnigan LTQ mass spectrometer operated at nominal mass resolution, which consisted of an electrospray ionization (ESI) source and linear ion-trap (LIT) mass analyzer. Raw data was extracted, peak-identified and QC processed using Metabolon's hardware and software. For studies spanning multiple days, a data normalization step was performed to correct variation resulting from instrument inter-day tuning differences. For every metabolite we computed the coefficient of variation (CV) of measurements by run day. We then took the median CV over run days and used that as a measure of the variability of the measurement process. Metabolites with a median CV greater than 0.25 or those where the CV could not be computed for at least two runs were excluded. In total we analysed 233 untargeted metabolites from 3029 individuals.

## Cis-QTLs discovery

Local SNP-phenotypes associations were performed for gene (FPKM), exon, splice phenotypes and protein levels using linear regression in FastQTL[60], with the seed number 1461167480. Principal component analysis (PCA) was used to control for unwanted technical variation. In addition, all analyses included sex, 3 PCs derived from genotype data, and a variable identifying the cohort of origin of the samples, called "center". Only SNPs in the region 1MB up- and down-stream of the TSS of each phenotype were considered for cis-QTLs. Raw read counts from exons and genes were used to discover cis-eQTLs after rank normalization, using a model that included 60 PCs for gene expression and 55 PCs for the exon level. Splicing-QTLs used rank normalized phenotypes generated with LeafCutter[58] and 20 PCs. Cis-pQTLs used rank normalized protein levels and 10 PCs. Missing protein measures, no more than 8.5% for any protein, were imputed using the mean of each protein. The coding genes for the proteins were used to center the cis-window for analysis. Two protein identifiers matched two genes: *FUT3* and *FUT5*. For simplicity, these were considered independent measurements during analyses.

Multiple testing correction was performed using a beta approximation coded in FastQTL. For exon-QTLs and splicing-QTLs, we employed the grouping strategy described in Delaneau et al. (--*grp* option)[55] to control for multiple phenotypes associated to the same TSS. This strategy considers all phenotype-SNPs pairs in a cis-window for a given gene at once, calculating a beta distribution per gene/TSS to assess their significance. The best phenotype-SNP per gene were reported as outcome in all analyses. To control the genome-wide false discovery rate (FDR), we used the qvalue[21] correction implemented in the software largeQvalue[61].

## Independent QTLs

Identification of secondary independent cis-QTLs was performed as described in Aguet et al.[12] and Viñuela et al.[3] using a stepwise regression procedure over all variants in the window using fastQTL at each stage fitting all other discovered signals as covariates in addition to the other covariates and PCs. This was done only on phenotypes with an QTL discovered in the association analysis (FDR < 1%). The maximum beta-adjusted *P* value (correcting for multiple testing across the SNPs and phenotypes) over these phenotypes was taken as the gene- or protein-level threshold. A cis-scan of the window was performed in each iteration using 1,000 permutations and correcting for all previously discovered SNPs. If this *P* value was significant, the best association was added to the list of discovered QTLs as an independent signal and the forward step proceeded to the next iteration. The backward stage consisted of testing each forward signal separately, controlling for all other discovered signals and covariates. The exon and splicing level cis-QTL scans used the -grp function and reported only the best association in each iteration.

## Trans-QTL discovery

Trans-QTL analysis was performed between molecular phenotypes with a genomic location (RPKM expression and proteins) and all SNPs further than 5MB from the TSS of the expressed gene or coding gene for the protein. For all associations, since phenotypes PCs may capture global trans effects removing true signals, we used residuals after removing known technical covariates with a linear mixed model implemented in the lme4 package in R[62]. To control for false positives in trans-eQTLs[57] we removed: any gene with low mappability and SNPs located in repetitive regions and with mapping issues. We then performed a genome wide scan of SNP-gene associations using QTLtools[55] storing all *P* values < 1e-04. Pairs of SNP-genes with known cross-mappability issues in a 100 kb window were then removed. Multiple testing corrections for trans-eQTLs was done using 50 permutations. The best *P* values per phenotype were used to define a phenotype level threshold and calculate adjusted *P* values for each gene. To adjust for multiple testing across genes, we used the qvalue package to estimate the false discovery rate. For proteins, 100 permutations per protein were used. For the identification of secondary trans-QTL, we used a stepwise regression analysis after defining non-overlapping cis-windows around 1MB up- and down-stream of any significant trans-QTLs signal. This created 2021 genomic regions for expression and 2589 regions for proteins to be tested for any nearby signal. Then, we ran the standard conditional analysis using FastQTL to look for further signals.

## Functional annotation

Functional enrichment analysis of eSNPs relative to pSNPs was done using available ChromHMM annotations from 14 blood cell

lines and VEP annotations[13,14]. Odds ratios for enrichment of cis-eQTLs relative to cis-pQTLs and cis-QTLs relative to trans-QTLs were computed in R.

## Transcription factor enrichment

To estimate the proportion of significant trans-SNPs which also affected local molecular phenotypes, we extracted all the cis-QTL $P$ values for SNPs with a significant trans-SNP effect and calculated $\pi_1$ estimates using all significant associations and the qvalue package. The product of all $P$ values per SNPs were used to calculate the probability of one gene to be associated to the trans-SNP, allowing us to calculate one $P$ value per phenotype using $\pi_1$ enrichment. To investigate if cis-SNPs that also have an effect in trans were more likely to regulate a transcription factor (TF) in cis, we extracted all genes or proteins reported by Lambert et al. [22] as transcription factors (TFs). We then used a Fisher exact test to calculate the OR of a cis-eQTL affecting TFs, given the number of TFs in the whole dataset that have trans associations.

## Metabolites-QTLs

Targeted and untargeted metabolites were residualised removing technical covariates using a linear mixed model. We performed a genome-wide association for each metabolite with no exclusion of any genomic region. Multiple testing correction was performed using 100 permutations per metabolite. Secondary signal identification used stepwise regression analysis after the identification of regions around primary associations.

## Tissue and phenotype specific associations, replications

To identify the proportion of cis-eQTLs, cis-pQTLs and metabo-QTLs active in other tissues, we extracted the same SNP-gene in GTEx v6p[25], GTEx v8[12], and InsPIRE[3] that were significantly associated in DIRECT. To identify the proportion of eQTLs that were identified in DIRECT as QTLs of any type, we extracted the SNP-gene pairs from the significant GTEx and Inspire significant associations. $P$ values were then evaluated for enrichment using qvalue. For metabolites-QTLs with no TSS of reference, we used all cis-eQTLs in 45 studies and cis-pQTLs in DIRECT associated to the lead metabo-SNPs to calculate $\pi_1$ estimates. To identify the proportion of trans-eQTLs and trans-pQTLs active as trans-eQTLs in other tissues we extracted all gene-SNP and protein-SNPs pairs from the trans-QTL significant results in DIRECT-blood in GTEx v8 data and performed a linear regression. We controlled for covariates reported by GTEx (PCs from genotypes 1 to 5, 15 to 60 inferred covariates depending on tissue, pcr, platform, and sex). We considered SNP-genes or SNP-proteins to be active in other tissues when a test for association in the GTEx v8 dataset was significant after multiple testing corrections (qvalue < 0.05).

Replication of distal QTLs was done by extracting $P$ values for genetic association from publicly available summary statistics. For trans-eQTLs we used associations from the eQTLGen consortium (https://eqtlgen.org/)[26], extracting all gene-SNP pairs. For trans-pQTLs we used associations from the INTERVAL consortium (http://www.phpc.cam.ac.uk/ceu/proteins/)[17]. In both cases we evaluated the replication level using π1 and the qvalue package across all pairs commonly present between studies. Replication of metabolites associations was performed using summary statistics from Long et al. [27], however only $P$ values < 1e-05 were available and therefore the degree of replication could not be reported. From the metabolites-SNP pairs matching between studies ($n = 65$), we considered those with Benjamin-Hochberg corrected $P$ values < 0.05 to be replicated.

## Causal inference

To identify pairs of QTLs, we selected all SNPs, or pairs of SNPs in high linkage disequilibrium ($R^2 > 0.9$), which were significantly associated with two phenotypes in cis or trans. We identified 23,539 cases with one genetic variant and two molecular phenotypes (simple trios), independently of the genomic location of the two phenotypes. To avoid bias due to quantification correlations, we removed all pairs involving expression phenotypes where at least one exon overlaps both genes. We then used Bayesian Networks (BNs) to learn the causal relationship between pairs of QTLs. We only considered network topologies that assume a causal effect from genetic variants towards molecular phenotypes, as the opposite effect does not have biological meaning. Three models were evaluated (Fig. 3): 1) Direction 1 model ([1] and [2]): The genetic variant affects first phenotype 1, then phenotype 2; 2) Direction 2 model ([3] and [4]): The genetic variant affects first phenotype 2, then phenotype 1; 3) Independent model ([5] and [6]): The genetic variant affects both phenotypes 1 and 2 independently. The probabilities can be described in the following formulas, for one or two SNPs:

1.  Direction 1, 1 SNP: P(Phenotype$_2$|Phenotype$_1$) P(Phenotype$_1$|SNP) P(SNP)
2.  Direction 1, 2 SNPs: P(Phenotype$_2$|Phenotype$_1$) P(Phenotype$_1$ | SNP$_1$) P(SNP$_1$)
3.  Direction 2, 1 SNP: P(Phenotype$_1$ |Phenotype$_2$) P(Phenotype$_2$|SNP) P(SNP)
4.  Direction 2, 2 SNPs: P(Phenotype$_1$|Phenotype$_2$) P(Phenotype$_2$ | SNP$_2$) P(SNP$_2$)
5.  Independent, 1 SNP: P(Phenotype$_1$ |SNP) P(Phenotype$_2$| SNP) P(SNP)
6.  Independent, 2 SNPs: P(Phenotype$_1$ | SNP$_1$) P(Phenotype$_2$ | SNP$_2$) P(SNP$_1$) P(SNP$_2$)

We used normalized quantifications after removing any technical covariates. In addition, we removed from the phenotypes the effects of other QTLs by including SNPs as covariates, creating two phenotypes per trio. Computation to generate these pseudo-phenotypes were done using the –single-signal option from CaVEMaN[63]. The phenotypes were used to compute the likelihood of the three possible BN topologies using the R/bnlearn package[64]. The best fit was evaluated using BIC score, with a trio considered for further evaluation if one of the three models had a BIC score >10 compared to each of the other two models.

## GWAS variants identification and colocalization

To identify genetic variants with known GWAS associations we used the GWAS catalogue v1.0.2, accessed 26/10/2020[30]. All SNPs with a QTL effect that were also lead GWAS variants by the catalogue were reported and used in further analysis. To provide a measure of the co-localization signals, we calculated the probability that a GWAS hit shares the same causal variant as a cis-eQTL by bayesian colocalisation analysis as implemented by COLOC[65] and in a subset of GWAS studies. We used GWAS summary statistics from 16 studies listed on Supplementary Data 15. SNPs were filtered to keep variants with $P$ values < 0.1, including all the variants in a 20 kb window around the lead cis-eQTL variant. The minor allele frequencies used for the analysis were those from the GWAS summary statistics. As cis-eQTL variants selected for theses analyses were previously found to be associated with the traits studied, we reported the probability that both GWAS and cis-eQTLs were shared as P(H4') = P(H4)/(P(H3) + P(H4)). H3 is the probability that both traits have different causal variants and H4 the probability of both traits sharing the same causal variant. For GWAS studies reporting different sample sizes per significant variant due to population differences, we calculated probabilities using the median number of samples across studies. To identify enrichment of GWAS associations among QTLs, we calculated and evaluated odd ratios using a fisher test. Since the GWAS catalog includes results from pQTL and metabolite-QTL studies, we removed any GWAS-SNP from blood proteins levels, metabolites levels and related phenotypes (Supplementary Data 18). The complete list of variants, SNPs and studies considered is included

in Supplementary Data 15. To evaluate if the enrichments were due to statistical power, we evaluated the odd ratios using two sets of randomly selected SNPs from the most significant associations eSNPs matching the number of pSNP ($N = 2027$) and metabolite-SNPs ($n = 236$) being evaluated.

## Networks construction

Networks were visualised using Cytoscape v3.7.2[66]. For all the QTLs results, a table was created including phenotype (target node), SNPs (source node) and type of association (interaction). The STRING enrichment function was used to evaluate the biological enrichment of some networks.

## Figures

Data and results figures were generated using R or Cytoscape[66] v3.7.2 for the networks. Figures 1A and 3A were partially created with BioRender.com. Figure 1B was created using a network from Cytoscape and a lolliplot figure generated using the lolliplot() function from the trackViewer package[67]. Figures were combined in panels using Inkscape v1.0.1. Tables and objects to load networks into Cytoscape are included in the Zenodo submission (https://zenodo.org/record/7521410).

## Reporting summary

Further information on research design is available in the Nature Portfolio Reporting Summary linked to this article.

## Data availability

The molecular and clinical raw data as well as the processed are available under restricted access due to the informed consent given by study participants, the various national ethical approvals for the present study, and the European General Data Protection Regulation (GDPR), individual-level clinical and molecular data cannot be transferred from the centralized IMI-DIRECT repository. Requests for access will be informed on how data can be accessed via the DIRECT secure analysis platform following submission of an appropriate application. The IMI-DIRECT data access policy is available at https://directdiabetes.org. As described in the methods section we used the human genome build GRCh37 as a reference for genomic location of genotypes and transcriptomics data, and Gencode v19[54] for gene models and TSS information (https://www.gencodegenes.org/human/release_19.html). For functional enrichment analyses we used dataset from the Ensembl Variant Effect Predictor[14] (VEP) information v98 (https://grch37.ensembl.org/info/docs/tools/vep/script/vep_download.html) and ChromHMM[13] models (https://egg2.wustl.edu/roadmap/data/byFileType/chromhmmSegmentations/ChmmModels/imputed12marks/jointModel/final/). GTEx v6p and v8 summary statistics were accessed using the GTEx Portal (https://www.gtexportal.org/home/). The 16 GWAS studies evaluated for co-localization and the links to their summary statistics are listed in Supplementary Data 15. Complete summary statistics including cis and trans genetic associations for gene expression, proteins and metabolites, as well as files to visualize networks on Cytoscape are freely available in the following link https://zenodo.org/record/7521410.

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

## Acknowledgements

We are grateful to IMI-DIRECT study participants who volunteered for phenotyping, and clinical and technical staff across involved European study centres who contributed to recruitment and clinical assessment of study participants. The work leading to this publication has received support from the Innovative Medicines Initiative Joint Undertaking under grant agreement 115317 (DIRECT), resources of which are composed of financial contribution from the European Union's Seventh Framework Programme (FP7/2007-2013) and EFPIA companies' in kind contribution. Information on the initiatives and activities of the IMI-DIRECT research consortium is available at https://directdiabetes.org.

## Author contributions

Conceptualization: A.B., E.T.D., A.V. Methodology: A.B., A.V. Software: A.B., A.V. Validation: A.S., A.V. Formal analysis: D.D., T.D., T.D.M., A.V. Investigation: R.C., M.H., C.H., A.G.J., T.K., Anu.M, And.M, T.J.M., F.P., D.P., V.R., L.R., F.R., P.W.F., G.F., M.W., Je.A., J.M.S., A.V. Resources: K.H.A., R.C., H.C., P.J.M.E., M.H., T.H., T.H.H., A.T.H., A.J.H., M.L., And.M,

T.J.M., D.M., M.M., P.B.M., F.P., V.R., M.R., F.R., H.T., L.T., J.V., H.V., O.P. Data curation: J.F.T., M.G.H., C.A.B., R.W.K., Jo.A, R.C., F.D.M., M.H., C.H., Anu.M, And.M, T.J.M., B.N., F.P., V.R., F.R., S.S., K.D.T., A.V. Writing-original draft: A.B., A.V. Writing- review & editing: A.B., G.N.G., F.R., J.M.S., E.R.P., E.T.D., A.V. Visualization: T.D., D.D., A.V. Supervision: A.B., B.J., E.R.P., E.T.D., A.V. Project administration: J.F.T., C.A.B., RWK, I.M.F., G.N.G., H.G., B.J., E.R.P., A.V. Funding acquisition: S.B., P.W.F., G.F., H.G., B.J., M.I.M., I.P., H.R., M.W., Je.A., J.M.S., E.R.P., E.T.D.

## Competing interests

S.B. has ownerships in Intomics A/S, Hoba Therapeutics Aps, Novo Nordisk A/S, Lundbeck A/S, and managing board memberships in Proscion A/S and Intomics A/S. As of June 2019, M.I.M is an employee of Genentech and a holder of Roche stock. E.P. has received honoraria from Sanofi and Lilly. The other authors declare no competing interests. E.T.D. is currently an employee of GSK. The work presented in this manuscript was performed before he joined GSK. All other authors declare no competing interests.

## Additional information

Andrew A. Brown [1], Juan J. Fernandez-Tajes[2], Mun-gwan Hong [3], Caroline A. Brorsson[4,5], Robert W. Koivula[6], David Davtian[1], Théo Dupuis[1], Ambra Sartori[7,8,9], Theodora-Dafni Michalettou [10], Ian M. Forgie[1], Jonathan Adam[11,12], Kristine H. Allin[13], Robert Caiazzo[14], Henna Cederberg[15], Federico De Masi[4], Petra J. M. Elders[16], Giuseppe N. Giordano[17], Mark Haid[18], Torben Hansen [13], Tue H. Hansen [13], Andrew T. Hattersley [19], Alison J. Heggie[20], Cédric Howald[7,8,9], Angus G. Jones[19], Tarja Kokkola[15], Markku Laakso[15], Anubha Mahajan[2], Andrea Mari [21], Timothy J. McDonald[22], Donna McEvoy[23], Miranda Mourby[24], Petra B. Musholt[25], Birgitte Nilsson[4], Francois Pattou[14], Deborah Penet[7,8,9], Violeta Raverdy[14], Martin Ridderstråle[26], Luciana Romano[7,8,9], Femke Rutters[27], Sapna Sharma[12,28], Harriet Teare[29], Leen 't Hart [27,30,31], Konstantinos D. Tsirigos[4], Jagadish Vangipurapu[15], Henrik Vestergaard [13,32], Søren Brunak[4,5], Paul W. Franks[17], Gary Frost [33], Harald Grallert[11,12], Bernd Jablonka[34], Mark I. McCarthy[2,65], Imre Pavo[35], Oluf Pedersen [36,37], Hartmut Ruetten[34], Mark Walker[38], The DIRECT Consortium*, Jerzy Adamski [39,40,41], Jochen M. Schwenk [3], Ewan R. Pearson [1], Emmanouil T. Dermitzakis [7,8,9,66] ✉ & Ana Viñuela [10,66] ✉

[1]Population Health and Genomics, Ninewells Hospital and Medical School, University of Dundee, Dundee DD1 9SY, United Kingdom. [2]Wellcome Trust Centre for Human Genetics, University of Oxford, Oxford OX3 7BN, United Kingdom. [3]Science for Life Laboratory, School of Biotechnology, KTH - Royal Institute of Technology, Solna SE-171 21, Sweden. [4]Department of Health Technology, Technical University of Denmark, Kongens Lyngby, Denmark. [5]Novo Nordisk Foundation Center for Protein Research, Faculty of Health and Medical Sciences, University of Copenhagen, Copenhagen DK-2100, Denmark. [6]Oxford Centre for Diabetes Endocrinology and Metabolism, University of Oxford, Oxford OX3 7LJ, United Kingdom. [7]Department of Genetic Medicine and Development, University of Geneva Medical School, Geneva 1211, Switzerland. [8]Institute for Genetics and Genomics in Geneva (iGE3), University of Geneva, Geneva 1211, Switzerland. [9]Swiss Institute of Bioinformatics, Geneva 1211, Switzerland. [10]Biosciences Institute, Faculty of Medical Sciences, University of Newcastle, Newcastle upon Tyne NE1 4EP, United Kingdom. [11]German Center for Diabetes Research (DZD), Neuherberg 85764, Germany. [12]Research Unit of Molecular Epidemiology, Institute of Epidemiology, German Research Center for Environmental Health, Helmholtz Zentrum München, Neuherberg 85764, Germany. [13]The Novo Nordisk Center for Basic Metabolic Research, Faculty of Health and Medical Science, University of Copenhagen, Copenhagen DK-2100, Denmark. [14]University of Lille, Inserm, Lille Pasteur Institute, Lille, France. [15]Internal Medicine, Institute of Clinical Medicine, University of Eastern Finland, Kuopio, Finland. [16]Department of General Practice, Amsterdam UMC- location Vumc, Amsterdam Public Health research institute, Amsterdam, The Netherlands. [17]Department of Clinical Science, Genetic and Molecular Epidemiology, Lund University Diabetes Centre, Malmö, Sweden. [18]Metabolomics and Proteomics Core, German Research Center for Environmental Health, Helmholtz Zentrum München, Neuherberg 85764, Germany. [19]Department of Clinical and Biomedical Sciences, University of Exeter College of Medicine & Health, Exeter EX25DW, United Kingdom. [20]Institute of Cellular Medicine, Faculty of Medical Sciences, Newcastle University, Newcastle upon Tyne, United Kingdom. [21]Institute of Neuroscience, National Research Council, Padova 35127, Italy. [22]Blood Sciences, Royal Devon and Exeter NHS Foundation Trust, Exeter EX2 5DW, United Kingdom. [23]Diabetes Research Network, Royal Victoria Infirmary, Newcastle upon Tyne, United Kingdom. [24]Nuffield Department of Population Health, Centre for Health, Law and Emerging Technologies (HeLEX), University of

Oxford, Oxford OX2 7DD, United Kingdom. [25]Global Development, Sanofi-Aventis Deutschland GmbH, Hoechst Industrial Park, Frankfurt am Main 65926, Germany. [26]Department of Clinical Science, Lund University, Malmö, Sweden. [27]Epidemiology and Data Science, VUMC, Amsterdam, The Netherlands. [28]Food Chemistry and Molecular and Sensory Science, Technical University of Munich, München, Germany. [29]Centre for Health Law and Emerging Technologies, Department of Population Health, University of Oxford, Old Road Campus, Oxford OX3 7DQ, United Kingdom. [30]Department of Cell and Chemical Biology, Leiden University Medical Center, Leiden, The Netherlands. [31]Department of Biomedical Data Sciences, Molecular Epidemiology section, Leiden University Medical Center, Leiden, The Netherlands. [32]Steno Diabetes Center Copenhagen, Copenhagen, Denmark. [33]Nutrition and Dietetics Research Group, Imperial College London, London SW7 2AZ, United Kingdom. [34]Sanofi Partnering, Sanofi-Aventis Deutschland GmbH, Frankfurt am Main 65926, Germany. [35]Eli Lilly Regional Operations Ges.m.b.H, Vienna 1030, Austria. [36]Center for Clinical Metabolic Research, Herlev and Gentofte University Hospital, Copenhagen, Denmark. [37]Novo Nordisk Foundation Center for Basic Metabolic Research, Faculty of Health and Medical Sciences, University of Copenhagen, Copenhagen DK-2100, Denmark. [38]Translational and Clinical Research Institute, Faculty of Medical Sciences, University of Newcastle, Newcastle upon Tyne, United Kingdom. [39]Department of Biochemistry, Yong Loo Lin School of Medicine, National University of Singapore, Singapore 117597, Singapore. [40]Institute of Experimental Genetics, German Research Center for Environmental Health, Helmholtz Zentrum München, Neuherberg 85764, Germany. [41]Institute of Biochemistry, Faculty of Medicine, University of Ljubljana, Ljubljana, Slovenia. [65]Present address: GENENTECH, 1 DNA Way, San Francisco, CA 94080, USA. [66]These authors contributed equally: Emmanouil T. Dermitzakis, Ana Viñuela. *A list of authors and their affiliations appears at the end of the paper. ✉e-mail: emmanouil.dermitzakis@unige.ch; ana.vinuela@newcastle.ac.uk

## The DIRECT Consortium

Andrew A. Brown [1], Juan J. Fernandez-Tajes [2], Mun-gwan Hong [3], Caroline A. Brorsson [4,5], Robert W. Koivula [6], David Davtian [1], Théo Dupuis [1], Theodora-Dafni Michalettou [10], Ian M. Forgie [1], Jonathan Adam [11,12], Kristine H. Allin [13], Robert Caiazzo [14], Henna Cederberg [15], Federico De Masi [4], Petra J. M. Elders [16], Giuseppe N. Giordano [17], Mark Haid [18], Torben Hansen [13], Tue H. Hansen [13], Andrew T. Hattersley [19], Alison J. Heggie [20], Cédric Howald [7,8,9], Angus G. Jones [19], Tarja Kokkola [15], Markku Laakso [15], Anubha Mahajan [2], Andrea Mari [21], Timothy J. McDonald [22], Donna McEvoy [23], Miranda Mourby [24], Petra B. Musholt [25], Birgitte Nilsson [4], Francois Pattou [14], Deborah Penet [7,8,9], Violeta Raverdy [14], Martin Ridderstråle [26], Luciana Romano [7,8,9], Femke Rutters [27], Sapna Sharma [12,28], Harriet Teare [29], Leen 't Hart [27,30,31], Konstantinos D. Tsirigos [4], Jagadish Vangipurapu [15], Henrik Vestergaard [13,32], Søren Brunak [4,5], Paul W. Franks [17], Gary Frost [33], Harald Grallert [11,12], Bernd Jablonka [34], Mark I. McCarthy [2,65], Imre Pavo [35], Oluf Pedersen [36,37], Hartmut Ruetten [34], Mark Walker [38], Jerzy Adamski [39,40,41], Jochen M. Schwenk [3], Ewan R. Pearson [1], Emmanouil T. Dermitzakis [7,8,9,66]✉, Ana Viñuela [10,66]✉, Kofi Adragni [42], Rosa Lundbye L. Allesøe [4], Anna A. Artati [12], Manimozhiyan Arumugam [13], Naeimeh Atabaki-Pasdar [6,17], Tania Baltauss [43], Karina Banasik [4], Anna L. Barnett [44], Patrick Baum [45], Jimmy D. Bell [46], Joline W. Beulens [27], Susanna B. Bianzano [47], Roberto Bizzotto [21], Amelie Bonnefond [14], Louise Cabrelli [48], Matilda Dale [3], Adem Y. Dawed [1], Nathalie de Preville [43], Koen F. Dekkers [30], Harshal A. Deshmukh [20], Christiane Dings [50], Louise Donnelly [1], Avirup Dutta [13], Beate Ehrhardt [51], Line Engelbrechtsen [13], Rebeca Eriksen [33], Yong Fan [13], Jorge Ferrer [52,53], Hugo Fitipaldi [17], Annemette Forman [13], Andreas Fritsche [54], Philippe Froguel [14], Johann Gassenhuber [34], Stephen Gough [6], Ulrike Graefe-Mody [55], Rolf Grempler [45], Lenka Groeneveld [27], Leif Groop [56], Valborg Gudmundsdóttir [4], Ramneek Gupta [37], Anita M. H. Hennige [57], Anita V. Hill [58], Reinhard W. Holl [59], Michelle Hudson [58], Ulrik Plesner Jacobsen [4], Christopher Jennison [51], Joachim Johansen [4], Anna Jonsson [13], Tugce Karaderi [4], Jane Kaye [24], Gwen Kennedy [60], Maria Klintenberg [61], Teemu Kuulasmaa [62], Thorsten Lehr [50], Heather Loftus [48], Agnete Troen T. Lundgaard [4], Gianluca Mazzoni [4], Nicky McRobert [6], Ian McVittie [23], Rachel Nice [22], Claudia Nicolay [63], Giel Nijpels [27], Colin N. Palmer [1], Helle K. Pedersen [4], Mandy H. Perry [22], Hugo Pomares-Millan [17], Cornelia P. Prehn [18], Anna Ramisch [7], Simon Rasmussen [4], Neil Robertson [6], Marianne Rodriquez [64], Peter Sackett [4], Nina Scherer [50], Nisha Shah [24], Iryna Sihinevich [50], Roderick C. Slieker [30], Nadja B. Sondertoft [13], Birgit Steckel-Hamann [42], Melissa K. Thomas [42], Cecilia Engel E. Thomas [37], Elizabeth Louise L. Thomas [46], Barbara Thorand [11,12], Claire E. Thorne [58], Joachim Tillner [34], Andrea Tura [21], Mathias Uhlen [49], Nienke van Leeuwen [30], Sabine van Oort [16], Helene Verkindt [14], Josef Vogt [13], Peter W. Wad Sackett [4], Agata Wesolowska-Andersen [6], Brandon Whitcher [46] & Margaret W. White [1]

[42]Lilly Research Laboratories, Eli Lilly and Company, Indianapolis, IN, USA. [43]Translational & Clinical Research, Metabolism Innovation Pole, Institut de Recherches Internationales Servier, Croissy sur Seine 78290, France. [44]Ninewells Hospital and Medical School, University of Dundee, Dundee DD1 9SY, United Kingdom. [45]Translational Medicine & Clinical Pharmacology, Boehringer Ingelheim International GmbH, Biberach an der Riss 88397, Germany. [46]Research Centre for Optimal Health, School of Life Sciences, University of Westminster, London, United Kingdom. [47]Therapeutic Area CardioMetabolism and Respiratory Medicine, Boehringer Ingelheim International GmbH, Biberach an der Riss 88397, Germany. [48]Clinical Research Centre, Ninewells Hospital and Medical School, University of Dundee, Dundee DD1 9SY, United Kingdom. [49]Science for Life Laboratory, School of Biotechnology, KTH - Royal Institute of Technology, Solna, Sweden. [50]Clinical Pharmacy, Saarland University, Saarbrücken 66123, Germany. [51]Department of Mathematical Sciences, University of Bath, Bath, United Kingdom. [52]Department of Metabolism, Digestion and Reproduction, Imperial College London, London, United Kingdom. [53]Regulatory genomics and diabetes, Centre for Genomic Regulation, Barcelona, Spain. [54]Medizinische Universitätsklinik Tübingen, Eberhard Karls Universität Tübingen, Tübingen, Germany. [55]Medicine Therapeutic Area Metabolism 1, Boehringer Ingelheim International GmbH, Biberach an der Riss 88397, Germany. [56]Department of Clinical Sciences, Diabetes & Endocrinology Unit, Lund University, Malmö, Sweden. [57]Boehringer Ingelheim International GmbH, Biberach an der Riss 88397, Germany. [58]NIHR Exeter Clinical Research Facility, Royal Devon and Exeter NHS Foundation Trust, Exeter, United Kingdom. [59]Institute for

Epidemiology and medical Biometry, University of Ulm, Ulm, Germany. [60]The Immunoassay Biomarker Core laboratory, School of Medicine, University of Dundee, Dundee, United Kingdom. [61]VO Endokrinologi, Enheten för diabetesstudier, University of Lund, Lund, Sweden. [62]Institute of Biomedicine, Bioinformatics Center, University of Eastern Finland, Kuopio, Finland. [63]Lilly Deutschland GmbH, Bad Homburg, Germany. [64]Biotech & Biomarkers Research Department, Institut de Recherches Internationales Servier, Croissy sur Seine 78290, France.

