## [Peer Review File · Nature Communications]

Genetic analysis of blood molecular phenotypes reveals common properties in the regulatory networks affecting complex traitsREVIEWER COMMENTS

Reviewer #1 (Remarks to the Author):

In this paper, the authors performed QTL analyses for the expression levels of genes, proteins, and metabolites from more than 3,000 subjects in the DIRECT consortium, evaluated shared QTLs between cis and trans regulations, regulations of genes, proteins, and metabolites, inferred causal relationships among SNPs and molecular phenotypes, and constructed genetic and molecular association networks. They also compared their results with those from other studies, e.g. GTEx, and tried to interpret GWAS results using their networks. One major contribution of this paper is the rich data from a large number of subjects, although this is somewhat tempered by the open access of the data. Extensive analyses were performed and carefully documented. I have the following comments that I would like the authors to address.

[1] The authors observed that when two expression traits shared an eQTL, the effect directions were opposite 30% of the times. Can the authors look more closely to see whether some genomic features are predictive of the concordance or discordance of the effect directions, in addition to the physical distance?

[2] The authors stated that 0% of trans-pSNPs had an effect on local protein levels (line 185). How confident is this estimate? The so-called pi1 methodology was used throughout the paper, can the authors also provide confidence intervals for their estimates?

[3] Results from GTEx v6p (line 520) were used for comparisons. How different will the general conclusions be for the v8 results?

[4] Bayesian networks were used to infer causal relationships among SNPs and molecular phenotypes. It is not clear how discretization was performed and how sensitive were the results to different discretization approaches.

[5] Five GWAS traits were considered. The GWAS results from these traits were lumped together. How much variations there are across different traits in terms of the usefulness of the networks? How about other classes of traits? How much can the networks help to fine map trait causing genes/variants? Can the authors perform a more comprehensive analysis instead of anecdotal discussions?

[6] Given the lack of access of individual level data, can the authors create a web portal where summary level data can be queried and downloaded?

Minor comments:

Line 267: "contra-intuitively" \diamond "counter-intuitively"

Line 294: maybe better to rephrase "found that those were" to "found that these SNPs were"

Reviewer #2 (Remarks to the Author):

The authors identified QTLs on gene expression, protein and metabolites using DIRECT cohort and conducted extensive analysis to evaluate their findings and pointed out a couple of examples of how the regulatory network they build can be useful to further understand biological mechanisms of complex traits. This is a great study given the sample size and the amount of analysis done for this manuscript. I believe the QTLs identified in this study will be attractive to many researchers and the resources will be very useful for many GWAS studies. However, it is almost disappointing that the

manuscript was very disorganized. I highly recommend proofreading more carefully before the next submission. I have pointed some typos and unclear sentences that I found below (minor comments). There were also some concerns in some of the analyses. Please see below for detailed comments.

Major comments

In the analysis of finding overlap of GWAS lead SNPs with QTLs (line 289), what is the distribution of diseases/traits of those 2828 SNPs? Are these SNPs enriched in specific diseases/traits or any domain of them? For example, are QTLs identified in this study is more enriched in disease/traits where molecular phenotypes in the blood is known to be involved in such as immunological diseases/traits or are they also overlap significantly with diseases/traits in which involvement of blood traits are not so obvious such as psychiatric disorders or metabolic traits.

Related to the first comment, why only 5 GWAS was selected to perform COLOC with QTLs? It would be more logical to me to colocate other diseases/traits from 2828 GWAS SNPs overlapping with QTLs, otherwise please justify.

The definition of colocalization is defined by $H4/(H3+H4)$. However, this can define pair or QTL and GWAS to be colocalized even if $H4$ and $H3$ are very small, for example, $H4=0.009$ and $H3=0.001$ can still end up with $H4/(H3+H4) = 0.9$. When $H0$, $H1$ or $H2$ has the highest posterior probability, that means at least one of QTL or GWAS has no genetic signal so it should not be considered to be colocalized. Shouldn't they be filtered by $H3+H4$ before identifying colocalization? Otherwise, please justify.

Minor comments

The claim that "Our results indicate that a sufficiently large sample size in blood can informative of regulation in other tissues" is a little misleading. Although the authors mentioned the tissue specific regulation may be missed afterward, it should be clarified that, QTL in blood can be only informative for genes, proteins or metabolites that are expressed in the corresponding tissues.

Throughout the manuscript, please use "gene expression" rather than just "expression" to indicate the expression of genes.

Line 122 "the" at the most right of the line is extra

Line 141 typo (or error while converting to PDF) "TSS (OR E044"

Line 147 "one SNP" instead of "one SNPs"

Line 156 "Genes with opposite effect eSNPs" should be "Genes that had eQTLs with opposite effect"

Line 192 "where" instead "were"

Line 195 "shared genetic regulation" instead of "the degree of sharing of genetic regulation"

Line 196 "first" is not necessary

Line 202 "had SNPs" instead of "had a SNPs"

Line 219 parenthesis is not closed

Line 226 "active in other tissues" ("in" is missing)

Line 303 "SNPs" instead "SNPS"

Line 305 "hereby" instead "here"?

Line 307 "than" instead "that"?

Line 333 typo "20:4n6"?

Line 351 "the lack of XXX reported previously" instead of "the reported lack of XXX"

Line 358 what is "potential GWAS effector transcripts"

Mixed use of Supplementary and Supplemental. Please use Supplementary throughout the manuscript.

Supplementary Data 14 is not referenced in the main text.

Supplementary Figure 16 is not referenced in the main text.

Line 314-315 928 SNPs are not provided in Supplementary Data 13

Reviewer #3 (Remarks to the Author):

In this manuscript Viñuela et al. have analyzed genotype, RNA-seq, proteomics and metabolite data from whole blood samples of 3,029 individuals of the DIRECT study. They built an extensive catalogue of cis- and trans-eQTLs, cis-sQTLs, cis- and trans-pQTLs and metabo-QTLs and provide valuable insights into the widespread allelic heterogeneity and pleiotropy that is enabled thanks to the highly powered sample size of the DIRECT study. Using Bayesian network analysis the authors tested causal networks of different molecular phenotype pairs for dependent and independent relationships. They further integrated molecular QTLs with GWAS data to construct networks of GWAS variants effects. The manuscript is well written, methods are well-described and the majority of performed analyses is meaningful and sound. However, most strikingly the authors do not cover any of the study-specific questions given that a population of pre-diabetic and newly diagnosed type 2 diabetes patients have been recruited (see below). Additional major points are the lack of replication of many QTL types (except for cis-eQTLs) and the missing link or explanation of the biological relevance of the findings in this manuscript. Major revisions would be needed but would also significantly increase the importance of the manuscript for the human genetics community.

Major points/general remarks:

1. Disease context is missing. Given that this is a disease cohort and one of the aims of the consortium is to identify diabetes biomarkers I am missing analyses that are addressing disease-specific questions such as: are there any QTLs that differ between healthy and this cohort? Are there any QTLs that differ between pre-diabetic and diabetic patients? How are the results presented in this manuscript related to diabetes etiology?
2. With blood not being the primary tissue of origin in regards to diabetes pathology can the author comment on the rationale of performing all these multiomics assays in blood samples (instead of other more relevant tissues) and add a corresponding section in the introduction and discussion part? Highlighting or reflecting the use of collecting easily accessible blood product samples in disease studies where the primary tissue is not blood would be extremely helpful for the science community. Also commenting on the question that "if most of the cis-eQTLs are replicated in GTEx and tissue-specific eQTLs can be found in publicly available, highly powered whole blood studies, is there a need to collect eQTL data in future disease cohorts?" would be a great addition in the discussion section.

3. Replication of trans-eQTLs, cis- and trans-pQTLs as well as metabo-QTLs are missing. Without any replication data it is difficult to assess if the downstream analyses (e.g allelic heterogeneity of trans-eQTLs, pQTL networks and intersection with GWAS variants) are meaningful or not.

4. Network analysis by itself without any descriptions of striking patterns, interpretations or additional analyses are a bit unsatisfactory leaving readers wonder what they are learning from these networks. Often times concluding sentences at the end of a paragraph are missing as well. Addressing these lacks would help to better understand the biological relevance of the results presented in this manuscript. Please find below a few examples:

a. Fig. 1B: Please specify why the pleiotropic network of POLR2J2&co were chosen for Fig.1B. Is this the one with the biggest network or any other rationale? Did the authors test if these genes (that share a pleiotropic eSNP) tend to be more highly coexpressed than genes with similar linear proximity to each other but without sharing a pleiotropic eSNP? What is the relationship of pleiotropic eSNPs and TADs. Understanding how pleiotropic effects can help to better understand genome architecture would strongly improve this manuscript.

b. Fig. S9 B: What do we learn from this highlighted network?

c. Fig. S7: can the authors comment on the huge trans-eQTL clusters shown on the top of the plot? What kind of genes are involved? Have they been described in previous (bigger) trans-eQTL data sets?

d. What is the hypothesis of exploring the degree of sharing of genetic regulation between cis-eQTLs and cis-pQTLs if they are not physiologically linked (whole blood eQTLs mainly capturing gene expression of immune cells and plasma pQTLs mainly capturing liver proteins)?

5. P.7 "Genes with opposite effect eSNPs were more likely to be further away from each other than those with the same direction of effect (Wilcoxon test Pvalue=7.16e-15).": Can the authors hypothesize what mechanism might be behind this observation? Are the opposite effects significant eQTL or similar to the vignettes in Fig. S.14 barely significant and resemble rather noise than real biology? If lots of non-significant opposite effects are included the whole section should be revised.

6. RETN trans-eQTL/ Fig. S25: Can p-values and effect size direction be added to the plot? The caption says that all three have the same direction of effect but it seems as if the effect of IKZF1 seems to be opposite and the effect on GRB10 barely significant. The way it's currently described it not convincing and this second trans-eSNP might need to be removed from the manuscript.

7. Fig. 2C:

a. Can the authors comment why spleen cis-eQTL seems to replicate particularly poorly compared to all other tissues?

b. Besides liver cis-eQTLs many other GTEx tissues (including brain tissues) show pi estimates of 100% in DIRECT pQTLs. How is this strong replication explained? Especially in the tissues such as brain where the gene product does not pass the blood-brain barrier? Assuming that in some of the highly replicated tissues only a small number of cis-eQTLs are tested, integrating the number of tested cis-eQTL per tissue (e.g. as dot size or in a separate table) would be helpful.

8. In general, while many of the supplementary figures are very content-rich, results of the main figures tend to rather show concepts or very generic/overview results with little content. I'd recommend to revise the overall figure structure (e.g. construct more dense and additional figure panels with results from the supplement).

a. Fig. 3: only shows the concept and the overall result. Somehow adding a vignette (e.g. boxplots of the most significant result) for each of the figure panels would be highly recommended

b. Fig. 4: the GWAS context is missing in the current figure. Can the GWAS traits be somehow incorporated?

Minor comments:

1. sQTLs have been mapped but are missing in most of the remaining analyses (network and GWAS analysis). Are there any particular reasons for this exclusion?

2. Full summary stats are only provided for cis-eQTLs but not for cis-sQTLs and cis-pQTLs. please add or explain why these are not provided

3. P.6 "pSNPs were enriched in 5' UTR variants relative to eSNPs (OR=2.84, Pvalue=8.6e-16), while eSNPs were enriched in active TSS (OR E044=4.95, Pvalue=7.7e-03)(Supplementary Figure 3, Supplementary Data 4).": Why was the active TSS enrichment chosen to be highlighted in the main

text if the enrichment in ZNF genes and repeats is much stronger? Please comment on the potential meaning of this strong ZNF enrichment as well.

4. Fig. 1B:

a. Add figure legend that explains all the symbols used in the figure (similar to Fig, 4)

b. Caption says "The upper diagram shows the network around the cis-window of POLR2J3." but isn't it rather the pink POLR2J2? please explain otherwise

c. Which chromosome is shown in the lollipop plot?

5. Fig. 2B-D: x-axis label should be corrected to "Sample size" instead of "Tissue"

6. Caption for Fig. 4C seems to be missing

7. Can the authors comment why individual technical covariate correction was applied instead of PEER correction for metabo-QTL mapping?

8. Fig. S1E:

a. x-axis numbers are missing.

b. the high bar on the left-hand side is quite striking. Can the authors comment on the characteristics of this group?

9. Fig. S12 D:

a. can the authors comment on the enrichment of high p-values (bar close to 1)? is this a statistical artefact?

b. Caption of panel D says "C" instead of "D"

10. Fig. S14: I'd remove this analysis and the corresponding section in the main text given that the opposite effect of all 4 examples is simply a non-significant effect where the opposite sign might just resemble stochasticity

11. Some supplementary figures seem to be missing figure panels (eg. Fig. S9,10,14)

12. Line 147: "associated to the expression [of] two or .."

13. Line 226: "Next, we investigated whether these cis-eSNPs active [in] other tissues were also regulating protein or metabolite levels "

14. Line 300: with smaller downstream consequences on GWAS traits [thethan] proteins or metabolites.

15. Line 507: "all Pvalues per SNPs we[re] used to calculate the probability of one gene to be associated to the trans-SNP, "

16. Multiple sections of the MS: bellow  below

Reviewer #1 (Remarks to the Author):

In this paper, the authors performed QTL analyses for the expression levels of genes, proteins, and metabolites from more than 3,000 subjects in the DIRECT consortium, evaluated shared QTLs between cis and trans regulations, regulations of genes, proteins, and metabolites, inferred causal relationships among SNPs and molecular phenotypes, and constructed genetic and molecular association networks. They also compared their results with those from other studies, e.g. GTEx, and tried to interpret GWAS results using their networks. One major contribution of this paper is the rich data from a large number of subjects, although this is somewhat tempered by the open access of the data. Extensive analyses were performed and carefully documented. I have the following comments that I would like the authors to address.

[1] The authors observed that when two expression traits shared an eQTL, the effect directions were opposite 30% of the times. Can the authors look more closely to see whether some genomic features are predictive of the concordance or discordance of the effect directions, in addition to the physical distance?

We followed the reviewer suggestion and revisited the list of cis-eQTLs with shared eSNPs. Firstly, we noticed a slightly higher proportion of genes sharing cis-eSNPs were located on different strands (58.94% of pairs). After evaluation, we found that pairs of genes sharing eSNPs with opposite direction of effects were more likely to be located on different strands of DNA compared to those with the same direction (OR = 1.52, Fisher test pvalue= 4.21e-06). Secondly, an enrichment analysis of SNPs associated to two or more genes with opposite direction of effects (N=583 eSNPs) using VEP compared to those eSNPs that had shared effects in the same direction found enrichment for “SNPs downstream genes” (New Supplementary Figure 2C, reproduced here). This suggests that the location of the effector allele with respect to the gene body, e.g.: up- or downstream the gene TSS, influences the effect it may have in regulating the expression of the genes. A similar analysis using ChromHMM annotations showed enrichment for SNPs with opposite directions of effect in categories such as “active enhancers” for multiple cell types from peripheral blood (OR=10.4, pvalue = 2.38e-03); while “transcription” and “transcription regulation” was enriched for SNPs with same direction of effect. These suggest those eSNPs that show the same direction of effects on multiple genes may have a more direct and stronger effect on expression by promoting transcription, while those with opposite effects may show weaker effects or effects mediated by other factors such as enhancer regulation.

We have incorporated these analyses and results to the main manuscript with the following text (Page 6, line 174 onwards):

Moreover, we observed that pairs of genes associated with a variant with opposite direction of effect were more likely to be further away from each other than those where the variant had the same direction of effect on both genes (Wilcoxon test Pvalue=7.16e-15), and were more likely to be located on different DNA strands (OR = 1.52, Fisher test pvalue= 4.21e-06). An enrichment analysis of SNPs associated to two or more genes with opposite direction of effects (N=583 eSNPs) using VEP found enrichment for “SNPs downstream genes” (Supplementary Figure 2C). This suggests that the location of the effect allele with respect to the gene body, e.g.: up- or downstream the gene TSS, influences the effect it may have in regulating the expression of the genes. The enrichment analysis using ChromHMM annotations found SNPs with opposite directions of effects were enriched in active enhancers for multiple cell types from peripheral blood (OR=10.4, pvalue = 2.38e-03); while SNPs with same direction of effect were enriched in regions classified as “transcription” and “transcription regulation” among others (Supplementary Figure 2D). These suggest those eSNPs with the same direction of effects on multiple genes may have a more direct

and stronger effect on expression by promoting transcription, while those with opposite effects may have effects mediated by other factors such as enhancer regulation. In summary, our results support previous reports of abundant pleiotropic effects on cis-eQTLs (Aguet and et al. 2020) with limited information for cis-pQTLs. Given the increased number of proteins and samples evaluated in pQTL studies and reports of overlapping genetic architecture properties with gene expression (Sun et al. 2018; Folkersen et al. 2020; Zhang et al. 2021), we expect these pleiotropic effects to be also abundant at the protein level.

[2] The authors stated that 0% of trans-pSNPs had an effect on local protein levels (line 185). How confident is this estimate? The so-called pi1 methodology was used throughout the paper, can the authors also provide confidence intervals for their estimates?

To calculate confidence intervals (CIs) for our pi1 estimates, we used bootstrap resampling to produce new samples from the Pvalue distribution of one type of molecular assay tested in another. As results, we can report that the 95% CI for the estimate of the number of trans-eSNPs that have an effect on local gene expression levels (pi1=77.34%) is [67.56%-82.02%]. This approach is more problematic when considering protein levels, as the small number of initial Pvalues means a limited number of bootstrap permutations that can be sampled. Pi1 was equal to 0 across all 1000 resamplings of Pvalues for protein effects of trans eSNPs, demonstrating the absence of any signal for these effects.

In light of these observations, we calculated CIs for all the cross tissue comparisons. The following figures shows the pi1 values for GTEx v8 cis-eQTLs active as DIRECT cis-eQTLs (A), GTEx v8 eQTLs as DIRECT cis-pQTLs (B) and GTEx v8 eQTLs as DIRECT metaboQTLs, all with CI values. As mentioned above, the small number of proteins available to be tested meant that the CIs produced can be unreliable for plot B (number of Pvalues ranging from 13 (kidney) to 311 (Thyroid)). In light of these results, we have now added the following text in the manuscripts regarding pi1 estimates for proteins:

- Page 9, line 262: *For cis-pQTLs, we observed π_1 estimates ranging from 0% (artery coronary tissue or minor salivary gland) to 94.9% (adipose visceral omentum)(Figure 2E). However, these estimates were calculated using only between 13 and 311 Pvalues, as the number of pQTLs was limited. These estimates therefore showed large confidence intervals (Supplementary Figure 5H-K), indicating a limited value to evaluate the level of activity of cis-eQTLs from multiple tissues acting as cis-pQTLs in blood.*

- Supplementary Figures: *1) Enrichment analysis of GTEx v8 eQTLs discovered as blood eQTLs in DIRECT dataset. Tissues are ordered by sample size from left (smallest) to right*

(largest). Horizontal bars indicate confidence interval (C.I.) calculated for π_1 estimates using bootstrap resampling and the Pvalue distribution of each type of molecular assay tested in another. J) Enrichment analysis of GTEx v8 eQTLs discovered as blood pQTLs in DIRECT dataset. C.I. were larger for this analyses and the number of overlapping proteins across tissues oscillates between 13 (kidney) and 311 (Thyroid). K) Enrichment analysis of GTEx v8 eQTLs discovered as blood metaboQTLs in DIRECT dataset.

[3] Results from GTEx v6p (line 520) were used for comparisons. How different will the general conclusions be for the v8 results?

Following the recommendation of this and other reviewers, we calculated estimates of shared genetic regulation across tissues and phenotypes using GTEx v8 (also used in the previous points), observing small changes in best/worst performing tissues. The ranges of shared genetic effects for cis-eQTLs across tissues changed from 89.7%-96.3% in v6p to 71.6%-91.2% in v8, indicating that the larger sample size and use of genotypes from whole genome sequencing increased the ability to find tissue specific eQTLs. We have also used the chance to explore how changes in the sample size of studies for expression in different tissues impact the conclusions of these types of analyses. This comparison is now included in the main manuscript with Figure 2D and Supplementary Figure 5H and reproduced here for the reviewer:

Page 9, line 258: *However, a comparison of these estimates with an earlier version of GTEx with a smaller sample size per tissue (v6p(GTEx Consortium 2017)), showed the increase in sample size for cis-eQTLs studies in less accessible tissues reduced the degree of shared genetic effects detected across tissues and phenotypes (Figure 2D, Supplementary Figures 5H).*

Figure 1E: Comparison of the π_1 enrichment analysis between an earlier version of GTE x (v6p) and a larger later version (v8) and the eQTLs found in DIRECT blood eQTLs. Only tissues shared across version of GTE x were included ($n=43$). Pancreatic islets is not a GTE x tissue and therefore the π_1 values did not change.

Supplementary Figure 5H: Change in π_1 estimates using GTE x v6p versus v8 eQTLs. The left plot shows the change in π_1 (y-axis) against the tissue sample size in v6p (x-axis), while the right plot shows the change of π_1 against the percentage of sample size increase between GTE x v6p and v8.

Bayesian networks were used to infer causal relationships among SNPs and molecular phenotypes. It is not clear how discretization was performed and how sensitive were the results to different discretization approaches.

No discretization was done. We tested three linear models using genotypes as a categorical variable, and treated molecular traits as continuous. For each tested graph (referred to as trios in the manuscript) we decomposed the likelihood into the sum of the likelihoods of each node conditional on nodes that had directed edges leading into them, and from this we calculated the BIC. Only when the difference in BIC values between one model and the other two was >10 did we consider that we had enough evidence to propose one causal model as the most likely model.

[5] Five GWAS traits were considered. The GWAS results from these traits were lumped together. How much variations there are across different traits in terms of the usefulness of the networks? How about other classes of traits? How much can the networks help to fine map trait causing genes/variants? Can the authors perform a more comprehensive analysis instead of anecdotal discussions?

We have extended the work around GWAS findings to provide a more comprehensive analysis. Firstly, we now provide a new Table 1 with a summary of the traits for which the lead GWAS variants was also a lead QTL in our analyses. From this table we observe that the most represented traits belong to blood related phenotypes, including blood cell counts, and proteins and metabolites circulating in blood or lipids. This is consistent with a network constructed on blood based data being more useful for blood related traits. Secondly, we have further evaluated the type of QTL-SNPs that were GWAS lead variants and found that a larger number of cis-eQTLs (n=2,445) compared to other types of QTLs (n<800). However, when looking at the proportion of discovered QTLs that were also GWAS hits, we observed an excess of metaboQTLs (63.12%). It is important to note that genetic associations with metabolites are generally reported as GWAS studies, and included in the GWAS catalogue as such, and therefore it is not a surprise to observe these large number of metaboQTLs among GWAS SNPs. On the other hand, trans-eQTLs are not reported in the GWAS catalogue, but we still observed that 34.48% were also GWAS variants. Thirdly, we have now added 11 more studies to the 2 blood counts and 3 lipid related GWAS evaluated using COLOC, including with other types of traits such as educational attainment, neuroticism, schizophrenia and cardiovascular diseases. A summary of the results and traits tested can be found in the new Figure 4A, with a complete list of results available in Supplemental Data 14.

Here we include the new figures and tables, as well as the new text (in blue) around the GWAS analysis:

Page 12, line 340 onwards: *The complete network of all connected genetic effects detected on genes, proteins and metabolites included 79,733 nodes (15,254 genes, 373 proteins, 172 metabolites and 63,795 SNPs) and 80,645 edges identifying significant QTLs, connected in clusters containing between 3 and 19,711 nodes. Nodes had an average of 4.31 edges connecting them to neighbouring nodes. To investigate how molecular phenotypes could have downstream consequences for the risk of disease, we extracted information from the GWAS catalogue (GWAS catalogue v1.0.2, accessed 26/10/2020 (Buniello et al. 2019)) and identified all SNPs that were lead GWAS variants and acted as QTLs in blood. In our network, we observed 2,828 GWAS variants (Table 1, Supplementary Data 13-14) connected with an average to 1.9 molecular phenotypes: in total 823 genes, 58 proteins and 44 metabolites were connected to GWAS variants. Among the traits more often observed to be associated to SNPs in the network, we found blood cell counts (33.01% of the 2,828 variants tested) and plasma metabolites or proteins levels (9.11%), suggesting blood related traits were better characterized by the network. We also investigated if blood related traits were more likely to co-localize with cis-eQTLs in blood than other phenotypes commonly studied using GWAS, using data from 16 different studies (Supplementary Table 14) that included blood-related traits such as lymphocyte and platelet counts (Astle et al. 2016), and other traits such as height (Yengo et al. 2018) and schizophrenia (Ripke, Walters, and O'Donovan 2020) (Methods). We found that 72.13% of all blood cell counts variants and 70.69% of all lipid traits variants co-localized with cis-eQTLs (COLOC probability > 0.9, Supplementary Data 16). However, we also identify a large number of co-localizing signals with other traits without a clear relationship with blood. For example, 77 of 97 variants (79.38%) associated to height (Yengo et al. 2018) had evidence of co-localization with eQTLs in blood, while we found 26 of 35 for Type 2 Diabetes (Mahajan, Wessel, et al. 2018). All in all, we identified thousands of candidate molecular phenotypes associated to GWAS traits in an accessible tissue, further analysis is required to investigate their potential causal role in disease given that QTL signals are often shared across tissues (Aguet and et al. 2020; Viñuela et al. 2020). Next, we evaluated the type of molecular phenotypes that may be involved in mediating the activity of GWAS variants. Using all SNPs reported as GWAS variants in the network, we found that these were connected to more molecular phenotypes than other variants in the network (Pvalue=6.7e-97, Wilcoxon test), and were more likely to be associated to proteins (OR=8.93, Pvalue=3.15e-25) or metabolites (OR=17.51, Pvalue=1e-09) than to gene expression (Supplementary Table 13). Per type of QTL, we observed that GWAS were more often cis- and trans-eQTLs (Figure 4A). However, when*

considering the number of tested phenotypes, we saw that a large percentage of SNPs involved in metaboQTLs were also GWAS variants (63.12%), followed by SNPs acting as trans-eQTLs (34.48%)(Figure 4A). This enrichment could be due to GWAS variants more likely acting via processes not captured by gene expression, such as post-transcriptional modification, or it could be due to lack of statistical power. As we have more power to identify eQTLs, a higher proportion of our gene expression associations represent weak biological effects, with smaller downstream consequences on GWAS traits than proteins or metabolites. To evaluate the influence of statistical power for different phenotypes, we repeated the enrichment considering only the most significant gene expression associations, matching the number of protein associated SNPs or the number of metabolite associated SNPs (Methods). While the relative metabolite enrichment remained significant (OR=2.99, Pvalue=1.51e-3), the protein enrichment was reversed (OR=10.9, Pvalue=3.28e-81 for gene expression over proteins), suggesting GWAS enrichment was here not driven by post transcriptional processes. Moreover, genetic associations with metabolites and proteins are generally reported as GWAS studies, and included in the GWAS catalogue, driving to some extent the large overlap between both. This is not true for trans-eQTLs signals, for which we observed a large percentage of trans-eSNPs also as GWAS variants for non-molecular traits and diseases. Overall, we observed that GWAS variants modulated the levels of more molecular phenotypes than non-GWAS variants associated to molecular phenotypes; in particular they were enriched in associations with metabolites and strong genetic effects on local and distal gene expression.

Trait	SNPs	Percentage total SNPs
Total Blood cell counts	1290	33.01
Metabolites/Proteins	356	9.11
Blood Pressure/CAD	338	8.65
Blood protein levels	248	8.77
BMI/Obesity	171	4.38
Lipids	133	3.4
Autoimmune/Inflammatory	129	3.3
Education/Brain/Behaviour	120	3.07
Height	106	2.71
ALL OTHER TRAITS	1513	38.62
Core binding factor acute myeloid leukemia	59	1.51
Heel bone mineral density	57	1.46
Type 2 diabetes	44	1.13
Multiple sclerosis	37	0.95
C-reactive protein levels	27	0.69

Table 1: Summary of the traits associated with GWAS variants that identify as QTLs in DIRECT whole blood and plasma molecular traits. We categorize traits based on similarity after observing a large proportion of GWAS variants were associated to blood cell counts, and circulating proteins and metabolites. A complete list of all traits (N=3,908) associated with GWAS variants (N=2,828) is included in Supplementary Data 13-15, percentages were calculated over total number of traits. In addition, we highlight 5 traits outside the summary categories with large number of SNPs overlapping with QTLs in the dataset, including core binding factor acute myeloid leukaemia (CBF-AML), a type of cancer affecting the normal haematopoietic process and T2D.

Trait/GWAS	Tested SNPs	SNPs with Pro>0.9	Percentage tested SNPs
Asthma	21	14	66.67
Blood pressure diastolic	20	13	65
Blood pressure systolic	20	12	60.00
BMI	73	58	79.45
CAD	6	2	33.33
Eczema	4	3	75.00
Educational attainment	23	19	82.6
HDL cholesterol	30	20	66.66
Height	97	77	79.38
LDL cholesterol	13	10	76.92
Lymphocyte count	36	31	86.11
Neuroticism	13	11	84.61
Platelet count	86	57	66.27
Schizophrenia	12	7	58.33
T2D	35	26	74.28
Triglycerides	15	11	73.33

Supplementary Table 16: We performed a co-localization analysis between signals identified by 16 GWAS studies and eQTLs from whole blood. We show the number of SNPs tested per study, as well as the number of those SNP with a probability of co-localization >0.9 , as defined by COLOC. We found evidence for co-localization ranging from 33% to 86% for SNPs associated with each GWAS.

Supplementary Figure 10A: Summary of the co-localization analysis results showing the distribution of probabilities for co-localization per trait, including all SNPs tested.

Figure 4A: Barplot showing the Number (left) of SNPs involved in QTLs that were also reported as lead GWAS variants by the GWAS catalogue. On the right, we show the percentage of QTL SNPs that were also reported as lead GWAS variants over the number of QTL-SNPs in each category of QTLs. Overall, we found the greatest overlap for cis-eSNPs, followed by trans-eSNPs. However, the highest proportion of overlap, considering the number of significant QTLs evaluated, was observed for metabo-SNPs, followed by trans-eSNPs.

[6] Given the lack of access of individual level data, can the authors create a web portal where summary level data can be queried and downloaded?

The clinical, sequence and molecular data are accessible. The only limitation imposed is that researchers are not allowed to download data into local machines as the participants' consent does

not allow researchers, including those involved in the DIRECT consortium, to do so. In the DIRECT web page, researchers can find information to request access to the computer hosting the data (<https://www.computerome.dk/>) to perform any further analyses.

Summary level data, on the other hand, has no access restrictions and was made freely available and downloadable for others to use in the repository Zenodo (<https://zenodo.org/record/4475681>). Since the release of the pre-print of this manuscript in medRxiv (ID: 21254347), files hosted there have been already downloaded 348 times (Nov-2022). In addition, we have received and fulfilled direct requests from other researchers for the complete summary statistics of the metabolites analysis, not only those p values $< 1e-04$. We are investigating how to make these files fully available as well.

Minor comments:

Line 267: “contra-intuitively” “counter-intuitively”

Line 294: maybe better to rephrase “found that those were” to “found that these SNPs were”

We have made these corrections in the text.

Reviewer #2 (Remarks to the Author):

The authors identified QTLs on gene expression, protein and metabolites using DIRECT cohort and conducted extensive analysis to evaluate their findings and pointed out a couple of examples of how the regulatory network they build can be useful to further understand biological mechanisms of complex traits. This is a great study given the sample size and the amount of analysis done for this manuscript. I believe the QTLs identified in this study will be attractive to many researchers and the resources will be very useful for many GWAS studies. However, it is almost disappointing that the manuscript was very disorganized. I highly recommend proofreading more carefully before the next submission. I have pointed some typos and unclear sentences that I found below (minor comments). There were also some concerns in some of the analyses. Please see below for detailed comments.

Major comments

1) In the analysis of finding overlap of GWAS lead SNPs with QTLs (line 289), what is the distribution of diseases/traits of those 2828 SNPs? Are these SNPs enriched in specific diseases/traits or any domain of them? For example, are QTLs identified in this study more enriched in disease/traits where molecular phenotypes in the blood is known to be involved in such as immunological diseases/traits or are they also overlap significantly with diseases/traits in which involvement of blood traits are not so obvious such as psychiatric disorders or metabolic traits.

We now provide a new Table 1 with a summary of the traits for which the lead GWAS variants was also a lead QTL in our analyses. From this table we observe that the most represented traits belong to blood related phenotypes, including blood cell counts, and proteins and metabolites circulating in blood or lipids. This is consistent with a network constructed on blood based data being more useful for blood related traits. We have also further evaluated the type of QTL-SNPs that were GWAS lead variants and found that a larger number of cis-eQTLs ($n=2,445$) compared to other types of QTLs ($n<800$). However, when looking at the proportion of discovered QTLs that were also GWAS hits, we observed an excess of metaboQTLs (63.12%). It is important to note that genetic associations with metabolites are generally reported as GWAS studies, and included in the GWAS catalogue as such, and therefore it is not a surprise to observe these large number of metaboQTLs among GWAS SNPs.

We have added the following text to the manuscript expanding this point:

Page 12, line 348: *Among the traits more often observed to be associated to SNPs in the network, we found blood cell counts (33.01% of the 2,828 variants tested) and plasma metabolites or proteins levels (9.11%), suggesting blood related traits were better characterized by the network. We also investigated if blood related traits were more likely to co-localize with cis-eQTLs in blood than other phenotypes commonly studied using GWAS, using data from 16 different studies (Supplementary Table 14) that included blood-related traits such as lymphocyte and platelet counts (Astle et al. 2016), and other traits such as height (Yengo et al. 2018) and schizophrenia (Ripke, Walters, and O'Donovan 2020) (Methods). We found that 72.13% of all blood cell counts variants and 70.69% of all lipid traits variants co-localized with cis-eQTLs (COLOC probability >0.9 , Supplementary Data 16, Supplementary Figure A10). However, we also identify a large number of co-localizing signals with other traits without a clear relationship with blood. For example, 77 of 97 variants (79.38%) associated to height (Yengo et al. 2018) had evidence of co-localization with eQTLs in blood, while we found 26 of 35 for Type 2 Diabetes (Mahajan, Wessel, et al. 2018). All in all, we identified thousands of candidate molecular phenotypes associated to GWAS traits in an accessible tissue, further analysis is required to investigate their potential causal role in disease given that QTL signals are often shared across tissues (Aguet and et al. 2020; Viñuela et al. 2020).*

Related to the first comment, why only 5 GWAS was selected to perform COLOC with QTLs? It would be more logical to me to colocalize other diseases/traits from 2828 GWAS SNPs overlapping with QTLs, otherwise please justify.

Colocalization analyses require access to full summary statistics, which often is not available or is computational demanding. Therefore, we chose to focus our efforts on traits observed in the list of 2828 overlapping GWAS signals, as the reviewer suggests, and that included lipid traits and blood counts. Additionally, we observed that the largest categories involved blood-related phenotypes, the same tissue of the molecular phenotypes, and wanted to explore a few examples in detail. However, and following the reviewers comments, we expanded the list of phenotypes evaluated to include traits not necessary driven by molecular activity in blood. The list now includes 16 studies with other types of traits such as educational attainment, neuroticism, schizophrenia or cardiovascular diseases. Additionally, we have included now a replication analysis for proteins and metabolites, traits included in the GWAS catalogue, unlike gene expression.

We have added text in the manuscript about the new co-localization phenotypes, copied in our previous answer, and about the replication of proteins and metabolites GWAS, which we include here:

Page 10, line 279 onwards: Next we assessed the ability of the DIRECT data set to identify distal genetic associations in other studies and tissues. Using GTEx v8, we tested an average of 1,662 gene-SNPs trans-eQTLs pairs per tissue (range = 1,512-1,760) and found that blood trans-eQTLs were also observed in whole blood ($\pi_1 = 0.336$) and brain putamen basal ganglia ($\pi_1 = 0.136$) among others, while 15 tissues did not identify any significant replicated trans-eQTLs from DIRECT blood ($\pi_1 = 0$, Supplementary Table 11). After multiple testing correction, we replicated 278 significant blood trans-eQTLs (237 unique genes, 159 unique SNPs), corresponding to 643 gene-SNP pairs across all tissues. In contrast, only 4 trans-pQTL were observed as trans-eQTLs in GTEx tissues after multiple testing correction: FCRL5-rs569841457 (adipose subcutaneous), MMP9-rs919377 adrenal gland, SPINK5-rs12462111 (nerve tibial) and OLR1-rs76604815 (uterus). Additionally, we evaluated the number of significant blood trans-eQTLs, trans-pQTLs and metabolite-QTLs that replicated in other blood and plasma datasets. Using the eQTLGen dataset (Vösa et al. 2021), we were able to evaluate 514 gene-SNP pairs from DIRECT trans-eQTLs, of which 463 were also significant (Supplementary Data 11). For cis and trans-pQTLs replication we used GWAS summary statistics from Sun et al. (Sun et al. 2018) and found that 281 cis-pQTL and 65 trans-pQTLs affecting 253 proteins replicated. For metabolites, we were able to evaluate 65 metabolite-SNPs pairs from 47 metabolites, of which all of them replicated in Long et al. (Long et al. 2017) (Supplementary Data 11).

The definition of colocalization is defined by $H_4/(H_3+H_4)$. However, this can define pair or QTL and GWAS to be colocalized even if H_4 and H_3 are very small, for example, $H_4=0.009$ and $H_3=0.001$ can still end up with $H_4/(H_3+H_4) = 0.9$. When H_0 , H_1 or H_2 has the highest posterior probability, that means at least one of QTL or GWAS has no genetic signal so it should not be considered to be colocalized. Shouldn't they be filtered by H_3+H_4 before identifying colocalization? Otherwise, please justify.

H_0 , H_1 and H_2 are probabilities that there is no genetic variant in the region that affects either the GWAS or the molecular trait, or both. The SNPs used were those already found to be associated with GWAS and the molecular phenotypes and therefore including these probabilities confuses discovery of genetic signals with testing for colocalisation. GWAS and molecular studies both have settled and well tested approaches for discovery of genetic signals that do not use COLOC, we prefer to use those for discovery keeping the two questions distinct. For this, we report the

probability of a shared signal, conditional on the fact that each trait has an already discovered associated genetic variant nearby.

Minor comments

The claim that “Our results indicate that a sufficiently large sample size in blood can be informative of regulation in other tissues” is a little misleading. Although the authors mentioned the tissue specific regulation may be missed afterward, it should be clarified that, QTL in blood can be only informative for genes, proteins or metabolites that are expressed in the corresponding tissues.

We agree that it should be clarified that the proportion of shared eQTLs is based only on genes expressed in both tissues and have added (in blue) the following clarifications:

Page 9, line 251: *Using Pvalue enrichment analysis (π_1) (Methods), we compared the distribution of Pvalues for significant cis-eQTLs for genes expressed across different tissues in DIRECT blood eQTLs, estimating that between 91.2% (pancreatic islets) and 71.6% (esophagus mucosa) of those cis-eQTLs were also active in whole blood (Figure 2B). Our results indicate that a sufficiently large sample size in blood can be informative of the regulation of genes expressed in other tissues, although a specific genetic process acting on specific genes, such as the effect of rs7903146 on TCF7L2 on pancreatic islets, may be missed if relevant tissues are not studied (Figure 2C, Supplementary Figures 5E-G).*

Throughout the manuscript, please use “gene expression” rather than just “expression” to indicate the expression of genes.

We have revised the manuscript adding the term when it was not clear.

Typos:

Line 122 “the” at the most right of the line is extra

Line 141 typo (or error while converting to PDF) “TSS (OR E044”

Line 147 “one SNP” instead of “one SNPs”

Line 156 “Genes with opposite effect eSNPs” should be “Genes that had eQTLs with opposite effect”

Line 192 “where” instead “were”

Line 195 “shared genetic regulation” instead of “the degree of sharing of genetic regulation”

Line 196 “first” is not necessary

Line 202 “had SNPs” instead of “had a SNPs”

Line 219 parenthesis is not closed

Line 226 “active in other tissues” (“in” is missing)

Line 303 “SNPs” instead “SNPS”

Line 305 “hereby” instead “here”?

Line 307 “than” instead “that”?

Line 333 typo “20:4n6”?

Line 351 “the lack of XXX reported previously” instead of “the reported lack of XXX”

Line 358 what is “potential GWAS effector transcripts” = This refers to molecules mediating GWAS activity. We have corrected the text now saying: “molecules mediating GWAS activity”.

We thank the reviewer for helping identify these typos. They all have been corrected and the text has been proofread.

Files to fix:

Mixed use of Supplementary and Supplemental. Please use Supplementary throughout the manuscript.

Supplementary Data 14 is not referenced in the main text.
Supplementary Figure 16 is not referenced in the main text.
Line 314-315 928 SNPs are not provided in Supplementary Data 13

We apologize for the oversight here and have now amended. Please note, that all files are also available in a public repository, to avoid changes due to formatting requirements from the journal (Zenodo: [10.5281/zenodo.4475681](https://zenodo.org/doi/10.5281/zenodo.4475681)).

Reviewer #3 (Remarks to the Author):

In this manuscript Viñuela et al. have analyzed genotype, RNA-seq, proteomics and metabolite data from whole blood samples of 3,029 individuals of the DIRECT study. They built an extensive catalogue of cis- and trans-eQTLs, cis-sQTLs, cis- and trans-pQTLs and metabo-QTLs and provide valuable insights into the widespread allelic heterogeneity and pleiotropy that is enabled thanks to the highly powered sample size of the DIRECT study. Using Bayesian network analysis the authors tested causal networks of different molecular phenotype pairs for dependent and independent relationships. They further integrated molecular QTLs with GWAS data to construct networks of GWAS variants effects.

The manuscript is well written, methods are well-described and the majority of performed analyses is meaningful and sound. However, most strikingly the authors do not cover any of the study-specific questions given that a population of pre-diabetic and newly diagnosed type 2 diabetes patients have been recruited (see below). Additional major points are the lack of replication of many QTL types (except for cis-eQTLs) and the missing link or explanation of the biological relevance of the findings in this manuscript. Major revisions would be needed but would also significantly increase the importance of the manuscript for the human genetics community.

Major points/general remarks:

1. Disease context is missing. Given that this is a disease cohort and one of the aims of the consortium is to identify diabetes biomarkers I am missing analyses that are addressing disease-specific questions such as: are there any QTLs that differ between healthy and this cohort? Are there any QTLs that differ between pre-diabetic and diabetic patients? How are the results presented in this manuscript related to diabetes etiology?

Following the reviewers request we have evaluated the molecular differences between pre-diabetics and diabetics by looking at how many of the genetic effects identified may show differences due to disease status. We re-calculated cis-eQTLs and cis-pQTLs separately for individuals diagnosed as diabetics and individuals diagnosed as pre-diabetics. Since one group is almost twice the size of the other, we randomly selected 900 samples from individuals classified as pre-diabetics to match the number of people with diabetes. The same individuals were used to calculate cis-eQTLs and cis-pQTLs. As shown on the following table, the number of cis-eQTLs and cis-pQTLs identified were very similar for the two groups. We see strong concordance in the Pvalues produced (Figure). In summary, we see little evidence that eQTL and pQTL signals differ in relation to diagnosis.

	cis-eQTLs	cis-pQTLs
Pre-diabetic samples	12393	283
Diabetic samples	11889	260

Figure: $-\log_{10}$ Pvalue for eQTLs from individuals with T2D and with preT2D status in the DIRECT dataset. In this figure the best gene-SNP pair in each of the dataset is plot, and therefore the same SNP may not be plotted.

Finally, we would like to note that the DIRECT consortium has already published 46 manuscripts, many of which evaluate the clinical and molecular differences between the pre-diabetics and diabetics individuals from this cohort, as well disease progression. The full link can be found in the consortium web (<https://directdiabetes.org/publications-to-date/>), and for a recent publication related to T2D, please see Allesøe et al, Nature Biotechnology (<https://doi.org/10.1038/s41587-022-01520-x>). The focus of our proposed project was genetic effects on molecular traits, and their consequences for disease related GWAS findings, a focus to which we stuck to avoid duplication of work and preserve consortium harmony. We and others are also currently further exploring and expanding this dataset to better understand the molecular regulatory consequences of disease development and hope to report soon with new analyses centred around the development of T2D.

2. With blood not being the primary tissue of origin in regards to diabetes pathology can the author comment on the rationale of performing all these multiomics assays in blood samples (instead of other more relevant tissues) and add a corresponding section in the introduction and discussion part? Highlighting or reflecting the use of collecting easily accessible blood product samples in disease studies where the primary tissue is not blood would be extremely helpful for the science community. Also commenting on the question that “if most of the cis-eQTLs are replicated in GTEx and tissue-specific eQTLs can be found in publicly available, highly powered whole blood studies, is there a need to collect eQTL data in future disease cohorts?” would be a great addition in the discussion section.

We very much agree with the reviewer that the message that whole blood can be used to better understand how disease genetics is important, with the advantage of much simplified sample collection. We have expanded multiple sections of the manuscript to further discuss the value of performing multi-omics analyses in accessible tissues. We have also included in the discussion an expanded paragraph on the advantages and limitations of using well powered molecular datasets on accessible tissues, vs smaller studies in more relevant disease tissues. The text is reproduced here:

Page 5, line 118 (Introduction): *An additional challenge for employing molecular phenotypes to identify the full causal relationship between genetic variants and complex traits is the availability of samples from the relevant disease tissue or cell type. Our own work has previously shown that using eQTL analyses to identify genes mediating GWAS activity benefits greatly from data in a disease relevant tissue (Viñuela et al. 2020). For example, it took a moderately large gene expression study in pancreatic islets to detect evidence that the gene TCF7L2 mediates the activity of the type 2 diabetes (T2D) loci with the same name. However, and in agreement with other publications (Aguet and et al. 2020; Grundberg et al. 2012), we also observed that many genetic effects are often shared across tissues, allowing to some extent the use of proxy tissues to study the genetics of common diseases. Given the practical difficulties of obtaining multi-omics datasets from non-accessible tissues, a question that remains unanswered is how deep molecular phenotyping in accessible tissues such as blood may aid in understanding the genetics of complex diseases.*

Page 15, line 438 (Discussion): *The choice of tissue for the study of genetic regulation is even more critical for those aiming to understand the underlying mechanism of GWAS variants effects on molecular phenotypes. A clear example is the TCF7L2 loci associated to T2D. Despite finding 6 independent cis-eQTLs for the gene, none of these involved the T2D loci, which has only been shown to have an effect on expression in pancreatic islets (Viñuela et al. 2020). Therefore, the tissue of choice to study complex traits remains critical for research involving the activity of specific variants. This is confounded with the difficulty of defining the relevant tissue for a given disease or trait (Gamazon et al. 2018; Ongen et al. 2017). First, such definitions imply a complex disease such as T2D or cardiovascular diseases are driven by genetic and environmental factors active only in one or a handful of tissues. However, we have been able to identify a large number of co-localizing QTLs with GWAS signals from traits without a clear relationship with blood. One reason,*

particularly when working with circulating proteins and metabolites, is those molecules may not have been produced in blood, reflecting genetic effects in non-accessible tissues, e.g.: CCL16. Other reasons include a large degree of sharing of genetic regulation across tissues, cross tissue regulation, and cross phenotype regulation. Until we learn the general principles of regulation of multiple molecular phenotypes and the possible coordinated impact of genetic variation in and across tissues, our knowledge about how genetic effects are cascade from the specific alleles to gene expression and other molecules to drive disease will remain limited. This is only currently possible using accessible tissues.

3. Replication of trans-eQTLs, cis- and trans-pQTLs as well as metabo-QTLs are missing. Without any replication data it is difficult to assess if the downstream analyses (e.g allelic heterogeneity of trans-eQTLs, pQTL networks and intersection with GWAS variants) are meaningful or not.

We have evaluated different available datasets for replication, all of which are also added as supplementary tables. The information has also been added to the manuscript with the following paragraph:

Page 10, line 279: *Next we assessed the ability of the DIRECT data set to identify distal genetic associations in other studies and tissues. Using GTEx v8, we tested an average of 1,662 gene-SNPs trans-eQTLs pairs per tissue (range = 1,512-1,760) and found that blood trans-eQTLs were also observed in whole blood ($\pi_1 = 0.336$) and brain putamen basal ganglia ($\pi_1 = 0.136$) among others, while 15 tissues did not identify any significant replicated trans-eQTLs from DIRECT blood ($\pi_1 = 0$, Supplementary Table 11). After multiple testing correction, we replicated 278 significant blood trans-eQTLs (237 unique genes, 159 unique SNPs), corresponding to 643 gene-SNP pairs across all tissues. In contrast, only 4 trans-pQTL were observed as trans-eQTLs in GTEx tissues after multiple testing correction: FCRL5-rs569841457 (adipose subcutaneous), MMP9-rs919377 adrenal gland, SPINK5-rs12462111 (nerve tibial) and OLR1-rs76604815 (uterus). Additionally, we evaluated the number of significant blood trans-eQTLs, trans-pQTLs and metabo-QTLs that replicated in other blood and plasma datasets. Using the eQTLGen dataset(Vösa et al. 2021), we were able to evaluate 514 gene-SNP pairs from DIRECT trans-eQTLs, of which 463 were also significant (Supplementary Data 11). For cis and trans-pQTLs replication we used GWAS summary statistics from Sun et al. (Sun et al. 2018) and found that 281 cis-pQTL and 65 trans-pQTLs affecting 253 proteins replicated. For metabolites, we were able to evaluate 65 metabolite-SNPs pairs from 47 metabolites, of which all of them replicated in Long et al. (Long et al. 2017) (Supplementary Data 11).*

4. Network analysis by itself without any descriptions of striking patterns, interpretations or additional analyses are a bit unsatisfactory leaving readers wonder what they are learning from these networks. Often times concluding sentences at the end of a paragraph are missing as well. Addressing these lacks would help to better understand the biological relevance of the results presented in this manuscript. Please find below a few examples:

a. Fig. 1B: Please specify why the pleiotropic network of POLR2J2 were chosen for Fig.1B. Is this the one with the biggest network or any other rationale? Did the authors test if these genes (that share a pleiotropic eSNP) tend to be more highly coexpressed than genes with similar linear proximity to each other but without sharing a pleiotropic eSNP? What is the relationship of pleiotropic eSNPs and TADs. Understanding how pleiotropic effects can help to better understand genome architecture would strongly improve this manuscript.

As the reviewer deduced, the network was selected for being the largest cis-network identified. This is now indicated in the figure legend. In addition, and following the reviewer's recommendation we proceed to annotate genes to TADS called in 8 different blood cell types (Javierre et al, 2016).

Across the different cell types, we found that on average 87% of the pleiotropic eSNPs were only associated with genes within the same TAD, while 13% were associated with genes in 2 different TADS. We found 2 SNPs that were associated with genes in 3 different TADS called with 2 of the 8 cell types in the first case, and only 1 cell type in the second. The results of this analysis has now been incorporated to the manuscript with the following text:

Page 6 line 171: To better understand how pleiotropic SNPs may be affecting gene expression, we annotate genes to topologically associated domains (TADs) called in 8 different blood cell types (Javierre et al. 2016). Across the different cell types, we found that on average 87% of the pleiotropic eSNPs were only associated with genes within the same TAD, while 13% were associated with genes in 2 different TADs (Supplementary Data 6).

b. Fig. S9 B: What do we learn from this highlighted network?

The network (now in Supplementary Figure 3D) show how some alleles affect different metabolites with opposite direction of effects, in a similar way as it was described for both proteins and gene expression. The figure legend now expands on this point, highlighting the consistency of biological effects observed across the three different molecular phenotypes considered in this manuscript.

c. Fig. S7: can the authors comment on molecules mediating GWAS activity in the huge trans-eQTL clusters shown on the top of the plot? What kind of genes are involved? Have they been described in previous (bigger) trans-eQTL data sets?

The large trans-eQTLs corresponds to the network around the trans-SNP rs1354034. As indicated in the manuscript, “this variant was associated with the expression of 297 genes and involved in platelet function regulation²⁸”. This trans-eQTL has been replicated multiple times and the SNP associated with platelet counts and function has also been replicated multiple times. In our full network is at the cluster of the big cluster discussed on page and presented around Supplementary Data 17. The locus and its local and distal effects has been previously discussed by Kolberg et al (eLife, 2020), a reference we are now including in the main text.

d. What is the hypothesis of exploring the degree of sharing of genetic regulation between cis-eQTLs and cis-pQTLs if they are not physiologically linked (whole blood eQTLs mainly capturing gene expression of immune cells and plasma pQTLs mainly capturing liver proteins)?

We expect plasma protein levels to reflect a mixture of circulating proteins from other tissues and proteins from blood cells and, as the reviewer suggests, we may not a priori expect genetic effects on circulating protein levels to be linked to genetic effects on whole blood expression. This is reflected by the degree of sharing we observe (0.65) which was lower than the sharing of GTEx eQTLs in DIRECT whole blood (though some of this difference may be due to technical reasons).

5. P.7 “Genes with opposite effect eSNPs were more likely to be further away from each other than those with the same direction of effect (Wilcoxon test Pvalue=7.16e-15).”: Can the authors hypothesize what mechanism might be behind this observation? Are the opposite effects significant eQTL or similar to the vignettes in Fig. S.14 barely significant and resemble rather noise than real biology? If lots of non-significant opposite effects are included the whole section should be revised.

All the eSNPs included in this comparison were associated at a genome wide significant level to two or more genes. Therefore, the number of false positives was controlled for and this cannot be driving our results on enrichment of opposite directions of effect. The example in figure S14 refers to SNPs significantly associated to both gene expression and the derived protein. Again, these associations were also genome-wide significant.

Following this and a request from reviewer #1, we have further evaluated the possible functional properties of shared eSNPs with opposite direction of effects. The enrichment did not provide any clear biological enriched function that would explain the observed effect. However, we found that pairs of genes sharing eSNPs with opposite direction effects were more likely to be located on different strands of DNA compared to those with the same direction of effect (OR = 1.52, Fisher test pvalue= 4.21e-06). This suggests that the location of the SNP with respect to the gene body due to the strand, e.g.: up- or downstream the gene TSS, could influence its regulation of expression. This has been experimentally observed with transcriptions activator-like effector (TALE) experiments [Uhde-Stone et al, 2014]. These TALEs has been shown to regulate cis expression when they bind to the sense strand of a gene. However, there is no clear mechanisms, or at least not one general mechanism that would explain all the cases we observe. We hope, therefore, that by highlighting our results in the manuscript, other researchers may be able to identify the possible mechanisms underlying these pleiotropic effects.

6. RETN trans-eQTL/ Fig. S25: Can p-values and effect size direction be added to the plot? The caption says that all three have the same direction of effect but it seems as if the effect of IKZF1 seems to be opposite and the effect on GRB10 barely significant. The way it's currently described it not convincing and this second trans-eSNP might need to be removed from the manuscript.

The reviewer is correct, we made a mistake and reported both cis-associations as having the same direction for *IKZF1* and *GRB10*, rather than opposite directions. which is indeed the case and reported in the summary statistics tables. We have now corrected the text to report this effect, while also reporting both pvalues and betas for clarity. We have also added a color code on the arrows to indicate whether the alternate allele increases or decreases the expression of the gene.

7. Fig. 2C:

a. Can the authors comment why spleen cis-eQTL seems to replicate particularly poorly compared to all other tissues?

It is difficult for us to say. The breadth of the GTEx tissues often means outlier tissues can crop up, and we do not know if differences are biological, technical (maybe due to the ability to extract mRNA or preserve it during the process) or have occurred by chance. Spleen is a tissue with small sample size, which means estimates are based on fewer Pvalues and thus more unstable.

b. Besides liver cis-eQTLs many other GTEx tissues (including brain tissues) show pi estimates of 100% in DIRECT pQTLs. How is this strong replication explained? Especially in the tissues such as brain where the gene product does not pass the blood-brain barrier? Assuming that in some of the highly replicated tissues only a small number of cis-eQTLs are tested, integrating the number of tested cis-eQTL per tissue (e.g. as dot size or in a separate table) would be helpful.

molecules mediating GWAS activity

There are two issues playing a role in these comparisons. Firstly, for GTEx tissues with smaller sample sizes, such as the brain tissues, the majority of the discovered eQTLs are located in the promoter. These tend not to be tissue specific and are more likely to affect transcription and translation, as reported by GTEx and other studies. With limited sample size only strong genetics effects are discovered, and these are more likely to be shared across tissues. This is supported by the fact that by using GTEx v8, all those pi1 estimates at 100% have disappeared. Secondly, we have now included CI for pi1 estimates to show the uncertainty in the estimates. These figures are now included as Supplementary Figure 5I-K.

8. In general, while many of the supplementary figures are very content-rich, results of the main figures tend to rather show concepts or very generic/overview results with little content. I'd recommend to revise the overall figure structure (e.g. construct more dense and additional figure panels with results from the supplement).

We have now re-arranged the figures according to the reviewer's advice. More data-content plots are included in larger panel figures. Following the journals requirements, we have also grouped supplementary figures in larger panels.

a. Fig. 3: only shows the concept and the overall result. Somehow adding a vignette (e.g. boxplots of the most significant result) for each of the figure panels would be highly recommended.

We have now added a new figure in the main panel 4 of the manuscript showing an example of a resolved trio for the genes *ANPEP* and *AP3S2*, both associated with rs11073891. We believe the examples show the logic behind the models. The figure and the legend are reproduced here:

4E) The casual network analysis supports a model where the downstream consequences of genetic variation are often mediated by other molecular phenotypes. F) Example for the model testing of the associations for rs11073891 and the gene expression of *AP3S2* and the expression of *ANPEP*. The first two plots show the cis-eQTL effects for both genes, with the alternative allele rs11073891_C reducing the expression of both genes. The results for the dependent model 1, testing if the effect of the SNP on *ANPEP* is mediated by *AP3S2*, show no changes in the associations and therefore no mediation effect. The dependent model 2, testing for the mediation of *ANPEP* in the SNP effect on *AP3S2* shows a change consistent with a mediation.

b. Fig. 4: the GWAS context is missing in the current figure. Can the GWAS traits be somehow incorporated?

We have now added two new plots to Figure 4 showing the categories of QTLs that include GWAS SNPs. In addition, we are now highlighting GWAS-SNPs with a distinctive colour in network graphs, highlight their relevance in the networks. The colour coding was included in the Cytoscape

object readers may download from Supplementary Data, allowing others to explore the connections of each of the ~2800 GWAS variants in details.

Minor comments:

1. sQTLs have been mapped but are missing in most of the remaining analyses (network and GWAS analysis). Are there any particular reasons for this exclusion?

There was not a deliberate exclusion from our part, we simply have limited space and many results. All QTL results were included in the network and evaluated in the GWAS analysis, including the sQTLs, but on space grounds we chose not to focus on these in the manuscript. We hope other researchers may want to follow up our analyses and further explore not only splicing but also other possible aspects of our results that we may have missed.

2. Full summary stats are only provided for cis-eQTLs but not for cis-sQTLs and cis-pQTLs. please add or explain why these are not provided.

We apologize for the error. All Pvalues are available in version 2 of the repository and are now freely accessible.

3. P.6 “pSNPs were enriched in 5’ UTR variants relative to eSNPs (OR=2.84, Pvalue=8.6e-16), while eSNPs were enriched in active TSS (OR E044=4.95, Pvalue=7.7e-03)(Supplementary Figure 3, Supplementary Data 4).”: Why was the active TSS enrichment chosen to be highlighted in the main text if the enrichment in ZNF genes and repeats is much stronger? Please comment on the potential meaning of this strong ZNF enrichment as well.

We have now added a reference to the ZNF enrichment in the main text.

Page 6, line 152: *However, functional enrichment analysis of eSNPs relative to pSNPs, using available ChromHMM annotations from 14 blood cell lines(Ernst and Kellis 2012) and VEP annotations(McLaren et al. 2016), found these two classes of regulatory variants had different properties. pSNPs were enriched in 5’ UTR variants relative to eSNPs (OR=2.84, Pvalue=8.6e-16) and in variants in zinc finger protein binding sites (up to OR=10.34, Pvalue=1e-3), a common DNA-binding motif involved in protein-DNA interactions.*

4. Fig. 1B:

- a. Add figure legend that explains all the symbols used in the figure (similar to Fig, 4)
- b. Caption says “The upper diagram shows the network around the cis-window of POLR2J3.” but isn’t it rather the pink POLR2J2? please explain otherwise
- c. Which chromosome is shown in the lollipop plot?

Yes, these were oversights. We have now corrected the caption of the figure and added a legend for clarity. The figure now also lists the chromosome in the figure.

5. Fig. 2B-D: x-axis label should be corrected to “Sample size” instead of “Tissue”.

We have corrected the axis.

6. Caption for Fig. 4C seems to be missing.

We apologize for this oversight. The caption is now included with the manuscript.

7. Can the authors comment why individual technical covariate correction was applied instead of PEER correction for metabo-QTL mapping?

Methods such as PEER or PCA correct for unknown covariates with a global effect. Since trans variants can effect many genes, these can be captured in PC/PEER factors and unsupervised correction methods can remove much of the true signal. For this reason PCA correction was only used for cis analyses, while technical correction was used for all trans analyses, including metabolites, trans-eQTLs and trans-pQTLs.

8. Fig. S1E:

a. x-axis numbers are missing.

b. the high bar on the left-hand side is quite striking. Can the authors comment on the characteristics of this group?

The high bar on the left-hand side corresponds to the number of eQTLs where the affected gene is the closest to the lead eSNP. In other words, there was no other transcription start site (TSS) from another gene in between the variant and the gene TSS. The high bar on the right is grouping all the cis-eQTLs (n=9,476) for which we found 20 or more genes in between the eGene and the eSNP, with a total range from 0 to 197 genes. This information is now included in the caption of the new figure with the axis corrected.

9. Fig. S12 D:

a. can the authors comment on the enrichment of high p-values (bar close to 1)? is this a statistical artefact?

b. Caption of panel D says "C" instead of "D"

The number of cases of trans-pQTLs that would be considered cis-pQTLs is too low (n=153) to identify any kind of enrichment. As a result, the distribution of probability is skewed to the right (near 1). This is discussed in the responses related the pi1 estimate and the difficulties of reliable values from small lists of Pvalues.

10. Fig. S14: I'd remove this analysis and the corresponding section in the main text given that the opposite effect of all 4 examples is simply a non-significant effect where the opposite sign might just resemble stochasticity

We are sorry not to have been clear on this, in all 4 cases the genetic effects are significant for both directions of effect and so it is not the case one has opposite direction due to a simple coin flip. We believe there is a wider context that means it is important to keep these results in the manuscript. We, and others before us, have observed these examples in different datasets of genetic effects with opposite direction of effects on gene expression and proteins. However, as they only seem anecdotal due to limitations in the number of proteins we measured and samples used in other studies, these cases are not reported in published manuscripts but are discussed in conferences or personal communications. Therefore, we believe is important to include them in this manuscript to encourage further reporting and research.

11. Some supplementary figures seem to be missing figure panels (eg. Fig. S9,10,14)

Apologies, these are the result of last minute changes to the figures included in the panels. We have now revised all Supplementary figures and captions.

The following typos were corrected as requested:

12. Line 147: “associated to the expression [of] two or ..”
13. Line 226: “Next, we investigated whether these cis-eSNPs active [in] other tissues were also regulating protein or metabolite levels “
14. Line 300: with smaller downstream consequences on GWAS traits [thethan] proteins or metabolites.
15. Line 507: “all Pvalues per SNPs we[re] used to calculate the probability of one gene to be associated to the trans-SNP, “
16. Multiple sections of the MS: bellow  below.

REVIEWERS' COMMENTS

Reviewer #1 (Remarks to the Author):

The authors have adequately addressed my previous comments, I have no further concerns.

Reviewer #2 (Remarks to the Author):

The authors have sufficiently answered my previous comments. I have no further comment.

Reviewer #3 (Remarks to the Author):

The authors have done substantial revisions, and the manuscript has improved significantly. All my previous remarks were addressed sufficiently. As a final comment, I would recommend adding the analysis done for Reviewer#3 remark 1 regarding disease context as supplementary figure. However, instead of p-value comparison (not an appropriate measure to examine context specificity) I would recommend using mashr (Urbut et al.). For readers who are interested in diabetes-specific genetic regulation of gene and protein expression it would be informative to see that there are hardly any diabetes-specific QTLs. I trust the authors to implement this suggestion appropriately.

REVIEWERS' COMMENTS

Reviewer #3 (Remarks to the Author):

Comment: The authors have done substantial revisions, and the manuscript has improved significantly. All my previous remarks were addressed sufficiently. As a final comment, I would recommend adding the analysis done for Reviewer#3 remark 1 regarding disease context as supplementary figure. However, instead of p-value comparison (not an appropriate measure to examine context specificity) I would recommend using mashr (Urbut et al.). For readers who are interested in diabetes-specific genetic regulation of gene and protein expression it would be informative to see that there are hardly any diabetes-specific QTLs. I trust the authors to implement this suggestion appropriately.

Response: *We have run mashr on the DIRECT data, but unfortunately we have found that this approach does not work with our dataset, with mashr reporting everything to be shared across contexts. We believe the reason for this is that mashr uses a Bayesian approach, with priors designed for studies with much lower sample sizes, that are not appropriate for our data. We would also add that this study is not designed to answer the question the reviewer raises: differences between patients and controls were deliberately minimised, with pre-diabetic individuals chosen to have HbA1c values near the diagnostic threshold. This means that even if there are diabetes specific QTLs, we are not powered to discover the difference, and may falsely conclude that they do not exist. This is supported by the figure mentioned by the reviewer, which show very similar pvalues for the analysis that attempted to identify those T2D specific QTLs.*